# Human CEACAM1 is targeted by a *Streptococcus pyogenes* adhesin implicated in puerperal sepsis pathogenesis

Erin A. Catton[1,14], Daniel A. Bonsor[2,3,14], Carolina Herrera[4], Margaretha Stålhammar-Carlemalm[5], Mykola Lyndin[6,7], Claire E. Turner ◉[8], Jo Soden[9], Jos A. G. van Strijp ◉[10], Bernhard B. Singer[7,15], Nina M. van Sorge ◉[10,11,12] ✉, Gunnar Lindahl ◉[5,13] ✉ & Alex J. McCarthy ◉[1,10] ✉

Life-threatening bacterial infections in women after childbirth, known as puerperal sepsis, resulted in classical epidemics and remain a global health problem. While outbreaks of puerperal sepsis have been ascribed to *Streptococcus pyogenes*, little is known about disease mechanisms. Here, we show that the bacterial R28 protein, which is epidemiologically associated with outbreaks of puerperal sepsis, specifically targets the human receptor CEACAM1. This interaction triggers events that would favor the development of puerperal sepsis, including adhesion to cervical cells, suppression of epithelial wound repair and subversion of innate immune responses. High-resolution structural analysis showed that an R28 domain with IgI3-like fold binds to the N-terminal domain of CEACAM1. Together, these findings demonstrate that a single adhesin-receptor interaction can drive the pathogenesis of bacterial sepsis and provide molecular insights into the pathogenesis of one of the most important infectious diseases in medical history.

Bacterial sepsis remains a major cause of disease in humans[1]. The sepsis that occurs in women after childbirth, puerperal sepsis, still causes the death of more than 75,000 women annually worldwide and is of particular concern in developing countries[2]. Puerperal sepsis is also of historical importance as epidemics swept through maternity wards in the eighteenth and nineteenth centuries and dramatically increased maternal morbidity and mortality[3–5]. The

cause of the epidemics was highly contentious and was only explained through pioneering work by Ignaz Semmelweis, who demonstrated that antiseptic techniques could prevent disease[4], and through the development of germ theory. Puerperal sepsis therefore represents a textbook example of a nosocomial infection. Evidence eventually accumulated that the epidemics were caused by the Gram-positive bacterium *Streptococcus pyogenes*, also known as group A

[1]Centre for Bacterial Resistance Biology, Section of Molecular Microbiology, Department of Infectious Diseases, Imperial College London, London SW7 2AZ, UK. [2]University of Maryland, Baltimore, MD 21201, USA. [3]NCI RAS Initiative, Cancer Research Technology Program, Frederick National Laboratory for Cancer Research, Frederick, MD, USA. [4]Section of Immunology of Infection, Department of Infectious Disease, Imperial College London, London W2 1NY, UK. [5]Department of Laboratory Medicine, Division of Medical Microbiology, Lund University, Lund 223 62, Sweden. [6]Sumy State University, Sumy 40000, Ukraine. [7]Institute of Anatomy, Medical Faculty, University of Duisburg-Essen, Essen 45147, Germany. [8]The School of Biosciences, The Florey Institute, The University of Sheffield, Sheffield S10 2TN, UK. [9]Retrogenix, Chinley, High Peak, SK23 6FJ Chinley, UK. [10]Department of Medical Microbiology, UMC Utrecht, Utrecht 3584 CX, The Netherlands. [11]Department of Medical Microbiology and Infection Prevention, Amsterdam UMC location University of Amsterdam, Amsterdam Institute for Infection and Immunity, Amsterdam 1105 AZ, The Netherlands. [12]Netherlands Reference Laboratory for Bacterial Meningitis, Amsterdam UMC, location AMC, Amsterdam 1105 AZ, The Netherlands. [13]Department of Chemistry, Division of Applied Microbiology, Lund University, Lund 221 00, Sweden. [14]These authors contributed equally: Erin A. Catton, Daniel A. Bonsor. [15]Deceased: Bernhard B. Singer. ✉e-mail: n.m.vansorge@amsterdamumc.nl; gunnar.lindahl@tmb.lth.se; a.mccarthy@imperial.ac.uk

*Streptococcus* (GAS)[5,6], but the molecular basis of this disease has remained unknown. Epidemiological studies have repeatedly indicated that an antigen designated R28[7,8], which has properties as an adhesin[9] and typically is found in *emm28* strains, is associated with *S. pyogenes* strains causing puerperal sepsis outbreaks[10–15]. A key question therefore arises: what is the role of R28 and does it allow *S. pyogenes* to overcome innate defense mechanisms to cause puerperal sepsis? This problem is of interest given the global increase in puerperal sepsis cases caused by *S. pyogenes*[2,16], and the lack of vaccines against this pathogen. Moreover, insights about puerperal sepsis caused by *S. pyogenes* may provide information relevant to the pathogenesis of puerperal sepsis caused by other bacterial pathogens.

The human immunomodulatory receptor CEACAM1 (carcinoembryonic antigen-related cell adhesion molecule 1) binds to some bacterial proteins that contain an Ig-like fold designated IgI3[17]. The CEACAM1 receptor, which belongs to the CEACAM family of proteins, is widely expressed on human epithelial, endothelial, and immune cells[18] and modulates cellular activities[19,20] (Fig. 1a). Here, we show that R28, which possesses an IgI3 domain, specifically targets CEACAM1 of human origin to promote events that would collectively favor the development of puerperal sepsis. These events include enhancing bacterial adhesion to epithelial cell barriers, interfering with epithelial wound healing, subverting the innate immune response, and promoting bacterial replication in blood. While it is classically assumed that a pathogen employs distinct molecular interactions to overcome each defense mechanism of the host, our data demonstrate that a single interaction can allow a pathogen to overcome multiple host defense barriers on the pathway to invasive disease.

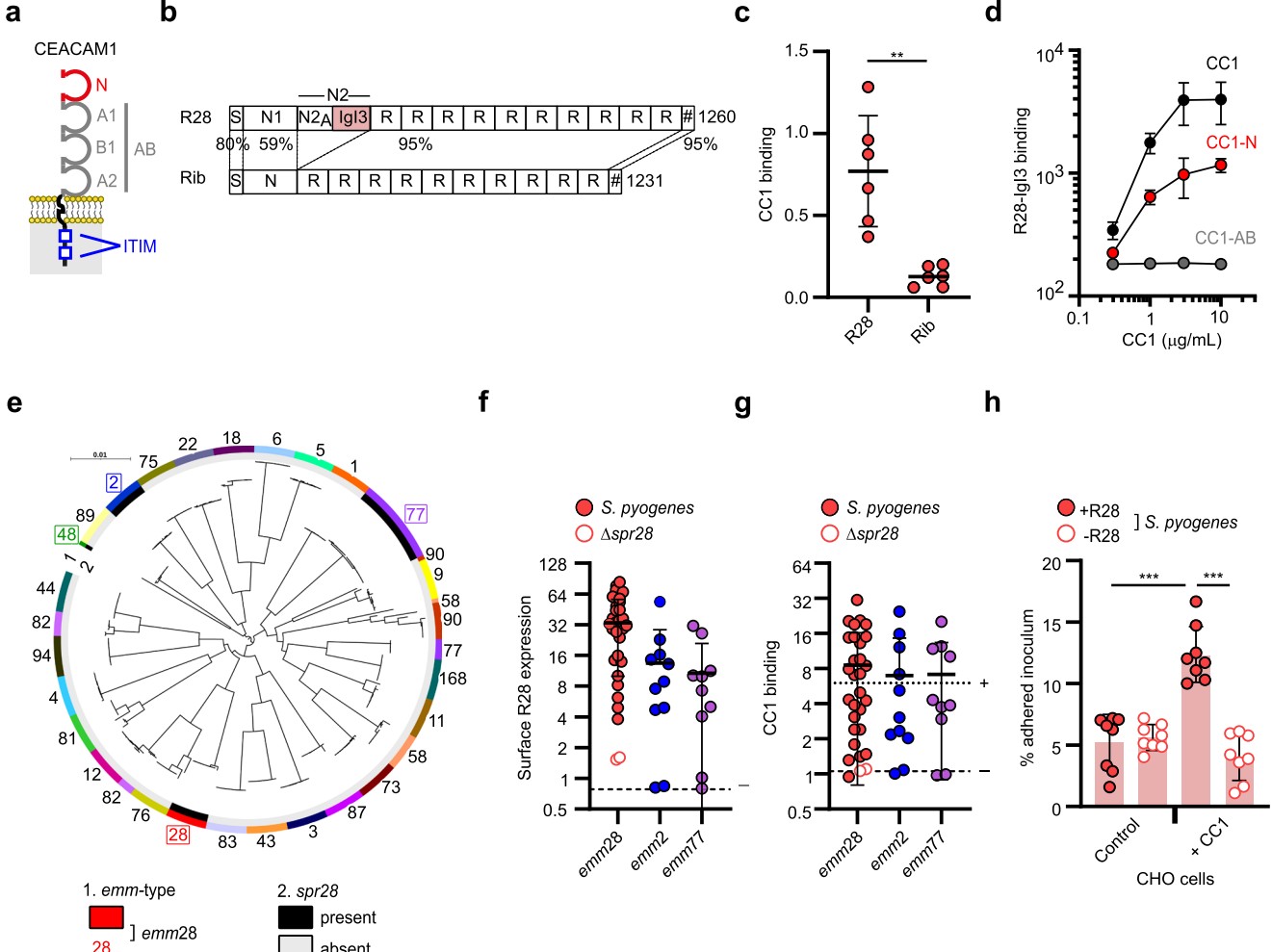

**Fig. 1 | The R28 adhesin of puerperal sepsis *S. pyogenes* isolates interacts with CEACAM1. a** Schematic of CEACAM1 (CC1), containing the extracellular region (composed of four Ig-like domains termed N, A1, B1, and A2) and cytoplasmic tail containing two immunoreceptor tyrosine-based inhibitory motifs (ITIM) for signaling. **b** Schematic of the *S. pyogenes* R28 and *S. agalactiae* Rib proteins. Each protein has a signal peptide (S) and cell wall anchor (#) domains. R28 contains the N2 domain that can be divided into $N2_A$ and $N2_B$ (renamed IgI3) subdomains. The percentage residue identity between domains is indicated. **c** Binding of recombinant (r)CC1-Fc to immobilized R28 and Rib, quantified by ELISA (mean ± s.d. of $n = 3$ independent experiments). Statistical significance tested by two-tailed paired Student's *t* test (**$p = 0.0079$). **d** Binding of rCC1-HIS variants to R28-IgI3, quantified by flow cytometry (mean ± s.d. of $n = 6$ independent experiments) **e** Core genome phylogeny of 287 representative *S. pyogenes* genomes. *spr28* positivity (inner circle) and *emm*-type (outer circle) given. Scale: substitutions per site. **f** Expression of surface-localized R28 by *S. pyogenes* strains, quantified by flow cytometry (Data are represented as mean ± s.d.). **g** Binding of rCC1-HIS to *S. pyogenes* strains as in **f**, quantified using flow cytometry (Data are represented as mean ± s.d.). In **f** and **g**, each data point represents the mean from $n = 2$ independent experiments per strain ($n = 30$ *emm*28 strains, $n = 11$ *emm*2 strains, $n = 10$ *emm*77 strains). Closed circles show *spr28*+ strains and open circles show isogenic Δ*spr28* strains. Open squares show *spr28*- strains, typed by PCR. Dashed lines indicate signal from an R28-negative strain (*S. pyogenes* M1 5448) or a CC1-binding strain (*S. agalactiae* A909). **h** Adherence of isogenic *S. pyogenes* AL368 strains to CHO cell lines (mean ± s.d. from $n = 8$ independent experiments). Statistical significance tested by one-way ANOVA with Tukey's post-hoc test (+R28 control vs. +R28 CC1 ***$p = 0.0006$, +R28 CC1 vs. -R28 CC1 ****$p = 0.0001$).

## Results

### *S. pyogenes* R28 adhesin binds CEACAM1

The R28 protein includes a long series of identical repeats and a unique N-terminal region[9], in which the IgI3 fold is contained in the N2 domain (Fig. 1b). We performed biochemical analysis of the interaction between intact R28 and CEACAM1 proteins, and found that recombinant (r)CEACAM1 bound to immobilized R28 isolated from the *S. pyogenes* cell wall, and also to β protein of *S. agalactiae* as previously described[17]. In contrast, rCEACAM1 did not bind to other purified streptococcal surface proteins tested including Rib of *S. agalactiae*, which is closely related to R28 but lacks the N2 part and the IgI3 region (Fig. 1b, c and Supplementary Fig. 1a)[9,21]. In a complementary test, purified R28 interacted directly with immobilized rCEACAM1 in a concentration-dependent manner (Supplementary Fig. 1b). The IgI3 domain of R28 was sufficient for the interaction (Supplementary Fig. 1c), and the N-terminal domain of CEACAM1 was necessary and sufficient for the interaction (Fig. 1d), as demonstrated in direct binding assays. In agreement with these findings, the binding of R28-IgI3 was specifically blocked by an antibody to CEACAM1-N (Supplementary Fig. 1c) and by the *Helicobacter pylori* protein HopQ, which binds to CEACAM1-N (Supplementary Fig. 1d)[22]. These data indicate that R28 and CEACAM1 interact through their IgI3 and N domains, respectively.

As R28 is strongly over-represented among *S. pyogenes* strains causing puerperal sepsis[8,10–12,14,15], it was of interest to analyze whether clinical *S. pyogenes* isolates interact with CEACAM1. We first analyzed the distribution of the gene (*spr28*) encoding R28 in whole genome sequences of 3886 *S. pyogenes* isolates representing 77 *emm*-types, and found that isolates belonging to *emm*2, *emm*28, and *emm*48 lineages, and one of two *emm*77 lineages, frequently carry the *spr28* gene (Fig. 1e). Among these lineages, isolates belonging to *emm*28 (serotype M28) are most common among strains causing outbreaks of puerperal sepsis[11]. Our analysis supports previous observations[13,23], and agrees with the finding that the *spr28* gene is located on a mobile genetic element[24]. Carriage of *spr28* gene, surface expression of the R28 protein and ability to bind CEACAM1 was evaluated for a large collection of strains belonging to *emm*28, *emm*2, or *emm*77 lineages, which included puerperal sepsis isolates. Of these, 28/28 *emm*28, 9/11 *emm*2, and 8/10 *emm*77 isolates carried *spr28*. Expression analysis (Supplementary Fig. 1e) showed that all strains carrying the *spr28* gene expressed R28, with highest expression in the *emm*28 strains (Fig. 1f), which included a puerperal sepsis isolate from the 1930s. Binding of CEACAM1 was observed for most of the *S. pyogenes* isolates (Fig. 1g, Supplementary Fig. 1f), and was concentration-dependent (Supplementary Fig. 1g). As expected, two *spr28*-negative *S. pyogenes* mutants lacked surface R28 expression and CEACAM1 binding (Fig. 1f, g and Supplementary Fig. 1e, f, h)[9]. The variation in CEACAM1 binding is likely due to differences in R28 expression, as supported by a positive correlation (Supplementary Fig. 1h), due to mutations in an intergenic region that impacts *spr28* transcription levels[25]. To analyze binding under more physiological conditions, we studied the ability of R28-expressing *S. pyogenes* to adhere to the surface of Chinese Hamster Ovary (CHO) cells and found that expression of human CEACAM1 enhanced the interaction through an R28-dependent mechanism (Fig. 1h and Supplementary Fig. 1i). These data demonstrate that R28-expressing *S. pyogenes* strains have an enhanced ability to adhere to epithelial cells through interaction with human CEACAM1.

### R28 binds CEACAM1 with high specificity

To investigate the specificity of CEACAM1 as a receptor for the R28 adhesin, we assessed whether R28 interacts with other human surface-expressed membrane proteins. High-throughput screening of the binding of R28 and the closely related Rib protein to 3359 cell line clones (derived from 2625 unique genes; Supplementary Data 1) was detected with rabbit anti-Rib serum (Fig. 2a), which detects both

R28 and Rib[9]. The screen revealed enhanced fluorescent signals for 45 cell lines compared to the control cell line (Fig. 2b). Further characterization of these hits revealed that the R28 protein, but not Rib protein, bound specifically to a CEACAM1-expressing cell line only (Fig. 2b).

Twelve CEACAM1 splice variants with different features can be expressed in human cells, including those that are membrane anchored or those that are soluble. They belong to a sub-family of CEACAM receptors, which share high sequence (~90% identity) and structural homology. However, CEACAM1-4L is the main expressed isoform and the only member containing the ITIM domains and subsequent immunomodulatory functions[18]. Because R28 did not bind to HEK293 cells expressing the CEACAM1 related receptors (CEACAM3, CEACAM5, CEACAM6, and CEACAM8; Supplementary Data 1), we tentatively concluded that R28 binds with high specificity to CEACAM1 on target cells. Biochemical and cellular assays were performed with purified preparations of R28 and the other CEACAM receptor family members. No interaction was detected between R28 and rCEACAM3, rCEACAM5, rCEACAM6 or rCEACAM8 in an ELISA (Fig. 2c) or in isothermal calorimetry assays (Supplementary Fig. 2). Additionally, R28-expressing *S. pyogenes* bacteria did not bind to rCEACAM3, rCEACAM5, rCEACAM6, or rCEACAM8 in flow cytometry assays (Fig. 2d). Finally, a significantly higher percentage of an R28-expressing *S. pyogenes* inoculum adhered to CEACAM1-expressing HeLa cells, as compared to HeLa cell lines expressing other CEACAMs (Fig. 2e). Taken together, these data reveal that CEACAM1 on epithelial cells is a highly specific receptor for the R28 adhesin of *S. pyogenes*.

### Structure of the R28-CEACAM1 complex

Structural information is essential, not only to understand how R28 targets its cognate receptor CEACAM1, but also to understand how R28 might be targeted by anti-virulence strategies. We solved the structure of R28-IgI3 bound to CEACAM1-N at a resolution of 3.05 Å (Fig. 3a; Supplementary Fig. 3a, b; and Supplementary Table 1). The asymmetric unit contains four copies of the (R28-IgI3)-(CEACAM1-N) complex (Supplementary Fig. 3a, the four paired molecules are A-B, C-D, E-F and G-H). Superposition of the complexes revealed little difference between them (r.m.s.d. 0.188–0.406 Å; Supplementary Table 2), and all analyses were performed using chains E and F (Supplementary Fig. 3b). The R28 domain has an Ig-like fold belonging to the IgI3 subset, which we recently identified as a domain in the β protein from *S. agalactiae* that binds CEACAM1 and CEACAM5[17]. Notably, R28-IgI3 has the following properties of the IgI3 fold: *A′* and *A″* strands, lack of *C′* and *C″* strands, and a truncated *C* strand (Supplementary Fig. 3c). A notable difference between the structure of the R28-IgI3 domain and the β-IgI3 domain is the presence in R28-IgI3 of a helix-turn-helix (HTH) motif between the *C* and *D* strands (Supplementary Fig. 3b–d).

We next assessed the mechanism of interaction between R28-IgI3 and CEACAM1-N. In the complex, R28-IgI3 targets the dimerization interface of CEACAM1-N (the *A′GFCC′C″* face) using a pocket of residues (K45, Y61, K64) that have high electrostatic potential (Fig. 3a; Supplementary Fig. 3e; and Supplementary Data 2). We further identified residues A55, E49, I52, I53, and I59 in R28-IgI3 as potential contact sites of CEACAM1-N (Fig. 3a; Supplementary Fig. 3f; and Supplementary Data 2) and tested alanine substitutions in these and additional R28-IgI3 residues to evaluate their importance in CEACAM1-N binding using biochemical assays. Significant reductions in binding of CEACAM1-N were observed for Dynabeads coated with R28-IgI3 mutated at K45, I52, I53, and Y61 (Fig. 3b and Supplementary Fig. 3g). These results were confirmed by ITC analysis (Supplementary Fig. 4a and Supplementary Table 3). Next, we tested whether alanine substitutions in CEACAM1-N residues altered R28-IgI3 binding (Fig. 3a, c and Supplementary Fig. 3f). Mutation of CEACAM1-N at residues F29, I91 or L95 abolished the interaction with R28-IgI3 (Fig. 3c), while mutation of CEACAM1-N at Q89 reduced binding to

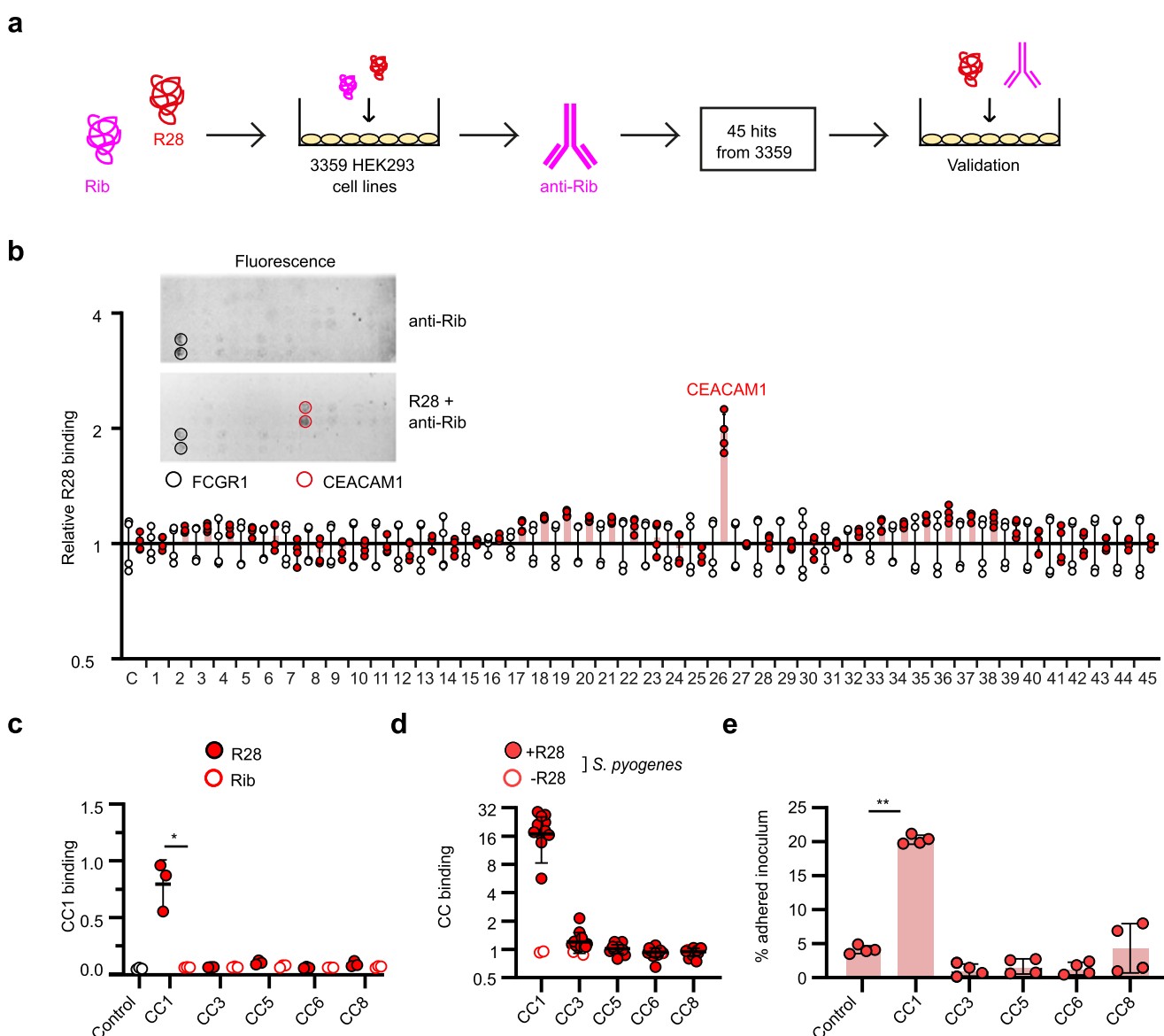

**Fig. 2 | CEACAM1 is a highly specific receptor for the R28 adhesin. a** Schematic of screening procedure for a library of 3359 HEK293 (Supplementary Data 1) cell line clones expressing human membrane proteins for ability to bind the streptococcal proteins R28 and Rib. Positive signals were obtained with 45 cell lines. **b** Binding of purified R28 protein to the 45 positively-hit HEK293 cell lines. Cells stained with rabbit anti-Rib sera (red) or normal rabbit sera (black) are shown. The relative fluorescence means ± s.d. of $n = 4$ from two independent experiments is shown. A representative image of fluorescence for the 45 cell lines (tested in duplicates) is shown in the inset, with FCGR1 identified as a non-specific hit. **c** Binding of recombinant CEACAM1-HIS (CC1), CEACAM3-HIS (CC3), CEACAM5-HIS (CC5), CEACAM6-HIS (CC6), and CEACAM8-HIS (CC8) to immobilized R28 and Rib

quantified by ELISA (Mean ± s.d. of $n = 3$ independent experiments). Statistical significance tested for binding of each rCC to R28 or Rib by paired two-sided Student's $t$ test (Rib + CC1 vs. R28 + CC1 $*p = 0.0266$). **d** Binding of rCC1-HIS, rCC3-HIS, rCC5-HIS, rCC6-HIS, and rCC8-HIS to a panel ($n = 15$, including two isogenic $\Delta spr28$ strains) of $S.$ $pyogenes$ strains from lineage $emm28$, quantified by flow cytometry (Each data point represents mean of $n = 3$ independent experiments). Mean ± s.d. of each lineage is shown. **e** Adherence of wildtype $S.$ $pyogenes$ AL368 to HeLa or HeLa$^{CC1}$, HeLa$^{CC3}$, HeLa$^{CC5}$, HeLa$^{CC6}$, and HeLa$^{CC8}$ cells (Mean ± s.d. of $n = 4$ independent experiments). Statistical significance compared to control cell line by one-way ANOVA with Dunnetts's $post$-$hoc$ test (HeLa control vs. HeLa.CC1 $***p = 0.002$).

R28-IgI3. Additionally, mutation of residues V96 or N97 significantly increased binding of CEACAM1-N to R28-IgI3. ITC analysis confirmed these findings (Supplementary Fig. 4b and Supplementary Data 3). Taken together, our structural and biochemical analysis indicates that R28-IgI3 docks over the CEACAM1 $cis/trans$ dimerization site of the $A'GFCC'C''$ face (residues F29, Q89, I91 and L95), and that the unique structural features of R28-IgI3 provide a site with high electrostatic charge (K45 and Y61) and high stability (I52 and I53) for CEACAM1 binding (Fig. 3d; Supplementary Fig. 3h). Finally, we investigated the presence of sequence variation in R28-IgI3 that could influence the capacity to interact with CEACAM1. The R28-IgI3

region was conserved in the genomes of those we tested for $spr28$ positivity (Fig. 1e) and BLAST analysis against the entire NCBI $S.$ $pyogenes$ protein database confirmed a high level of conservation with just a single variant of valine to alanine at amino acid residue 26. This residue is not located in or near the CEACAM1 binding interface suggestive that all R28 proteins interact with CEACAM1.

As R28-IgI3 binds to the $trans$-dimerization interface (the $A'GFCC'C''$ face) of CEACAM-N, we questioned whether the binding of R28-IgI3 to CEACAM1 could prevent CEACAM1 $trans$-dimerization interactions. Superimposition of the crystallographic CEACAM1 $trans$ dimer (PDB: 4WHD) with the R28-IgI3 complex shows that R28-IgI3

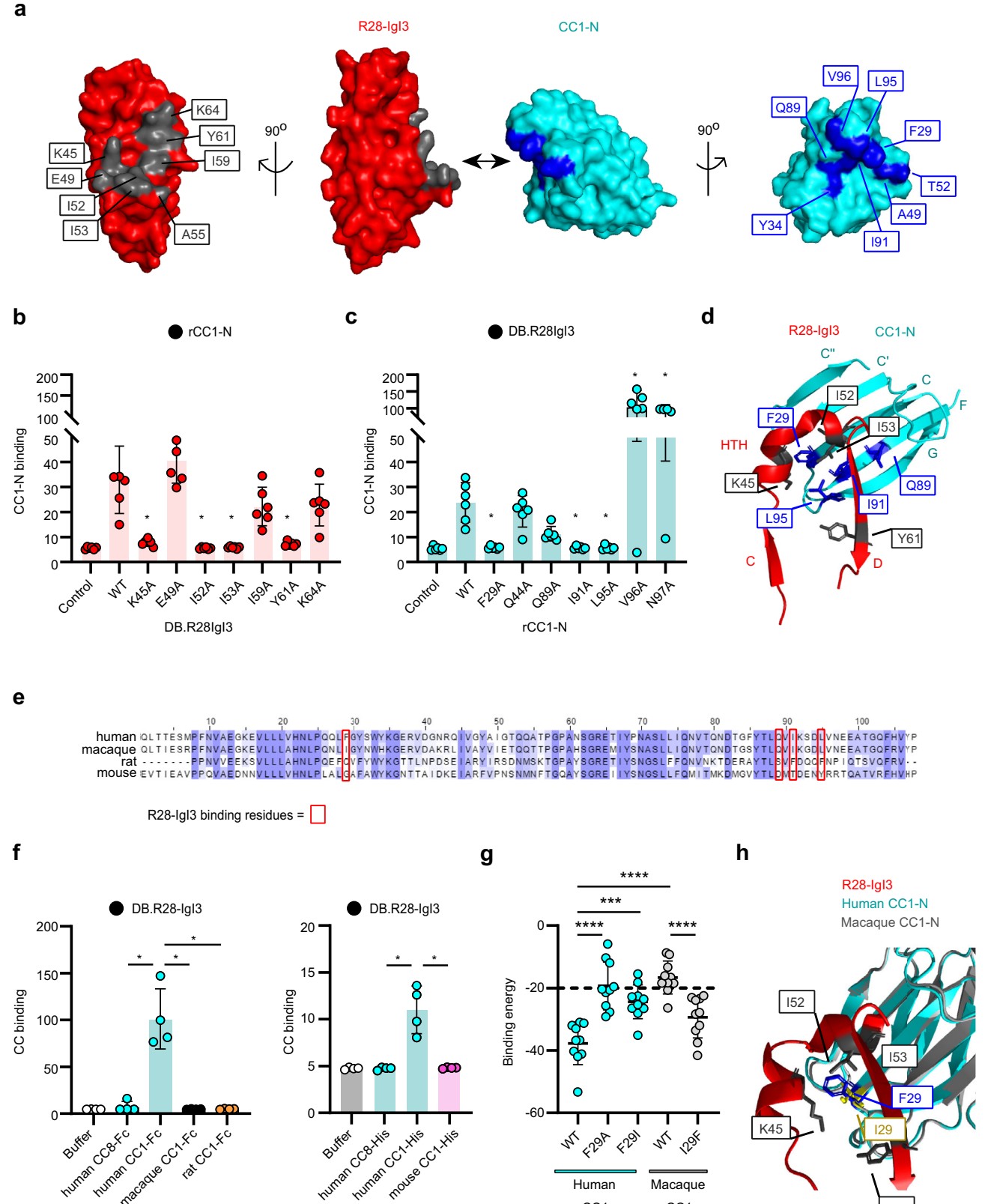

binding is incompatible with CEACAM1 *trans*-dimerization (Supplementary Fig. 5a, b). Similarly, the binding of other bacterial ligands is incompatible with CEACAM1 *trans*-dimerization (Supplementary Fig. 5c, d, e)[26]. These data suggest that infection with R28 + *S. pyogenes*, but not R28- *S. pyogenes*, would reduce cross-linking efficiency of CEACAM1.

## R28 specifically binds to human CEACAM1

*S. pyogenes* infections are human restricted. Nonetheless, the virulence properties of R28-expressing *S. pyogenes* strains have been studied in murine and non-human primate (macaque) models[9,25,27]. We therefore analyzed the host-specificity of the R28-CEACAM1 interaction. The four critical residues of human CEACAM1 (F29, Q89, I91, L95) required

**Fig. 3 | Structural basis and human specificity of R28-IgI3 interaction with CEACAM1. a** Crystal structure of N-terminal domain of CEACAM1 (CC1) domain in complex with R28-IgI3, shown as an electron density map. **b** Mutation of residues K45, I52, I53 and Y61 in R28-IgI3 abolishes binding to CC1-N (Mean ± s.d. of $n=6$ independent experiments). Statistical significance tested between R28-IgI3 wild-type and variants by one-way ANOVA with Dunnetts's *post-hoc* test (WT vs. K45A *$p = 0.0219$, WT vs. I52A *$p = 0.0210$, WT vs. I53A *$p = 0.0214$, WT vs. Y61A *$p = 0.0257$). **c** Mutation of residues F29, I91 and L95 in CC1-N abolishes binding to R28-IgI3 (Mean ± s.d. of $n=6$ independent experiments). Statistical significance tested between CC1-N wildtype and variants by one-way ANOVA with Dunnetts's *post-hoc* test (WT vs. F29A *$p = 0.0134$, WT vs. I91A *$p = 0.0122$, WT vs. L95A *$p = 0.0152$, WT vs. V96A *$p = 0.0471$, WT vs. N97A *$p = 0.0428$). **d** A close-up view of the R28-IgI3 binding interface of CC1-N. **e** Alignment of CC1-N sequences. The critical residues required for binding of R28-IgI3 to human CC1 are highlighted by red boxes. **f** Binding of human, macaque, mouse or rat rCC1-HIS or rCC1-Fc proteins to R28-IgI3 (Mean ± s.d. of $n=4$ independent experiments). Statistical significance tested by one-way ANOVA with Tukey's *post-hoc* test (Fc-tag proteins, human CC1 vs. human CC8 *$p = 0.0478$, human CC1 vs. macaque CC1 *$p = 0.0354$, human CC1 vs. rat CC1 *$p = 0.0353$; HIS-tag proteins, human CC1 vs. human CC8 *$p = 0.0489$, human C1 vs. mouse CC1 *$p = 0.0487$). **g** Docking of R28-IgI3 onto human and macaque CC1-N variants, quantified as the binding-free energy (kcal/mol) (Mean ± s.d. of $n=10$ in silico simulations). Binding-free energy for non-interacting proteins (R28-IgI3 and human CC1-N$^{F29A}$) is shown by a dashed line. Statistical significance tested by one-way ANOVA with Tukey's *post-hoc* test (human CC1 vs. human CC1$^{F29A}$ ****$p < 0.0001$, human CC1 vs. human CC1$^{F29I}$ ***$p < 0.0003$, human CC1 vs. macaque CC1 ****$p < 0.0001$, macaque CC1 vs. macaque CC1I29F ***$p = 0.0005$). **h** Close-up of the R28-IgI3 binding interface of human and macaque CC1-N. R28-IgI3 residues required are shown in gray. F29 in human CC1-N and I29 in macaque CC1-N are shown in blue and yellow, respectively.

for R28-IgI3 docking are not conserved in the macaque, mouse, and rat CEACAM1 sequences (Fig. 3e), which suggests that the R28-CEACAM1 interaction is human restricted. Biochemical assays revealed that R28-IgI3 bound to human CEACAM1 but not mouse, rat or macaque CEACAM1 (Fig. 3f). In agreement, R28-expressing *S. pyogenes* bacteria did not interact with macaque CEACAM1 in molecular and cellular assays (Supplementary Fig. 6a, b).

Human and macaque CEACAM1 only differed at one of the four residues shown to be critical for the R28-IgI3 interaction (residue F29 in human CEACAM1). Simulated docking of R28-IgI3 onto wildtype and mutated CEACAM1-N domain structures (Supplementary Fig. 6c) correctly predicted that R28-IgI3 binds to human, but not macaque CEACAM1 (Fig. 3g). Moreover, in silico analysis also revealed that an F29I mutation in human CEACAM1 significantly reduced the R28-IgI3 binding energy, while an I29F mutation in macaque CEACAM1 significantly increases the R28-IgI3 binding energy (Fig. 3g). Thus, the analysis indicated that the F29 residue in human CEACAM1 is a critical determinant of human-specificity, likely through interaction with residues K45 and I52 on R28-IgI3 (Fig. 3h). These results show that R28 specifically binds to human CEACAM1 and indicate that results from experimental animal models likely underestimate the role of R28 in *S. pyogenes* virulence.

## R28 impairs wound repair

As CEACAM1 is expressed on female reproductive tract epithelia (Supplementary Fig. 7a)[18,28], we tested the hypothesis that R28 enhances the adhesion of *S. pyogenes* to these epithelial surfaces through engagement of CEACAM1. Adhesion of our reference *S. pyogenes* strain AL368 to monolayers of the ME-180 cell line, a human cell line of cervical origin that expresses CEACAM1 (Supplementary Fig. 7b, c)[29], was dependent on R28[9] and was specifically reduced by an antibody that recognizes the N-domain of CEACAM1 (Fig. 4a). Similar observations were made for an alternative isogenic pair of *S. pyogenes* strains (Supplementary Fig. 7d). Thus, R28 promotes binding of *S. pyogenes* to human cervical cells by binding CEACAM1. Similar results were obtained in assays measuring the binding of fluorescently labeled *S. pyogenes* to detached ME-180 cells (Supplementary Fig. 7e, f, g).

The results obtained with the ME-180 cell line demonstrated that R28 interacts with the CEACAM1 receptor present on the surface of cervical epithelial cells. However, to cause puerperal sepsis, *S. pyogenes* must enter female reproductive tract tissue through the wounds generated during childbirth, subvert the innate immune response and resist killing by innate immune cells upon entry into the blood. We speculated that R28 was involved in some of these processes. To analyze the contribution of the R28-CEACAM1 interaction to the entry of *S. pyogenes* into tissues through wounds, we employed an in vitro scratch assay that is routinely used to investigate wound healing[30]. Notably, previous results had demonstrated that CEACAM1 regulates

cell migration and wound repair[31–33], although it has never been proposed that *S. pyogenes* (or R28) could modulate this process. Scratches inflicted to ME-180 cell monolayers closed by 40.1% (±11.7) after 24 h of culture and by 66.8% (±14.0) after 48 h of culture (Fig. 4b, c), but repair of the wound was strongly delayed after infection with R28-expressing *S. pyogenes*, while the R28-negative mutant had little or no effect. Furthermore, blocking access to the N-domain of CEACAM1 on the ME-180 cells restored the repair of the wound upon *S. pyogenes* infection (Supplementary Fig. 8a, b). Wound repair was also delayed in an R28-dependent manner upon infection with an alternative isogenic pair *S. pyogenes* (Supplementary Fig. 8c). To understand whether the R28-CEACAM1 interaction was sufficient to induce the delayed wound repair phenotype in the absence of other bacterial components, we investigated the capacity of purified R28 to delay wound repair. Repair of the wounds in ME-180 cell layers was delayed by incubation with purified R28, but not the closely related Rib protein of *S. agalactiae* (Fig. 4d). The phenotype was also elicited by rR28-IgI3 (Fig. 4e), indicating that the IgI3 domain interaction with ME-180 cells was sufficient to delay wound repair. To investigate whether the rR28-IgI3 domain acted through a CEACAM1-dependent mechanism, we compared the functional properties of rR28-IgI3 variants that bind or do not bind CEACAM1 (as revealed in Fig. 3b). Importantly, CEACAM1 binding rR28-IgI3 proteins (rR28-IgI3 and rR28-IgI3$^{E49A}$) delayed wound repair, whilst the CEACAM1 non-binding rR28-IgI3$^{Y61A}$ did not delay wound repair (Fig. 4e). Though there was not a significant difference in wound repair between rR28-IgI3 and rR28-IgI3$^{I53A}$, there was a trend. Collectively, these data demonstrate that interaction between *S. pyogenes* and human CEACAM1 through R28 promotes adhesion to epithelial cells and suppresses wound repair.

## R28 suppresses innate immune responses

The next defense mechanisms that *S. pyogenes* must pass is the innate immune response of a variety of immune cells that reside within the epithelium and lamina propria. Since CEACAM1 is an immunomodulatory receptor expressed on immune cells that regulates anti-bacterial responses[19,20], we hypothesized that expression of R28 by *S. pyogenes* would impair the anti-streptococcal immune response of tissues. To test this, we assessed the cytokine and chemokine profiles of wounded ME-180 cell monolayers and underlying human peripheral blood mononuclear cells (PBMCs) using transwell culture chambers. Non-dividing *S. pyogenes* strains were added to the apical chamber and an interaction was allowed to progress for 72 h prior to analysis of cytokine and chemokine production (Supplementary Fig. 9a). The immune response against wildtype *S. pyogenes* was impaired in comparison to that raised against the isogenic R28-negative strain (Supplementary Fig. 9b). Notably, there was a significant reduction, or trend for reduction, of the production of proinflammatory cytokines and chemokines including IL-1β, TNF-α, GM-CSF, IFN-γ, RANTES, IL-6, and MCP-2. To further investigate the immunomodulatory effect of R28 in

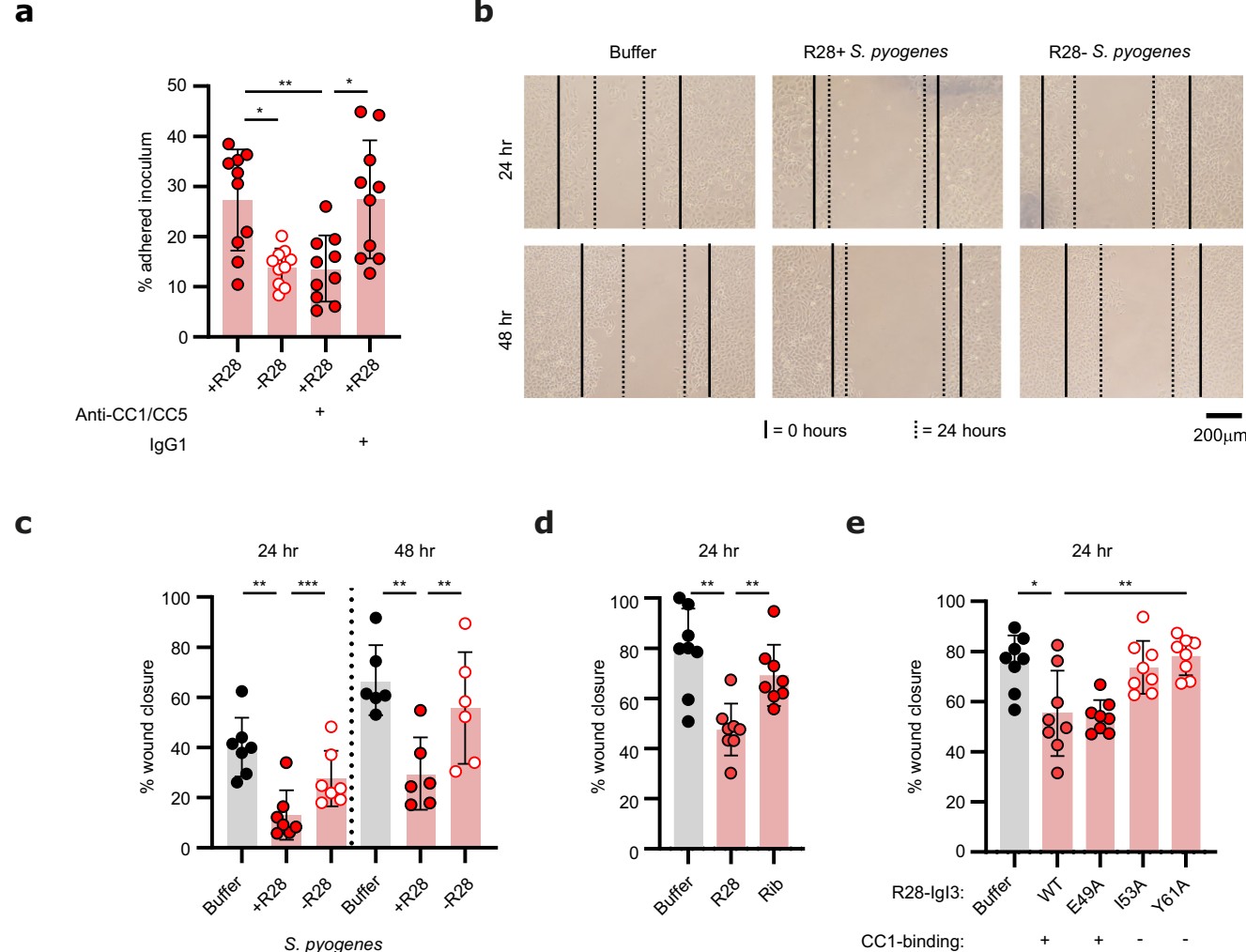

**Fig. 4 | R28 promotes adherence of *S. pyogenes* to human cervical epithelial cells and suppresses wound closure through a CEACAM1-dependent mechanism. a** Adherence of isogenic *S. pyogenes* AL368 strains to ME-180 cells at a multiplicity of infection (MOI) of 10, in the presence or absence a monoclonal antibody (mAb) specific to human CC1 and CC5 N-terminal (mean ± s.d. of *n* = 10 independent experiments). Statistical significance tested by one-way ANOVA with Tukey's *post-hoc* test (+R28 vs. −R28 *p* = 0.0142, +R28 vs. +R28 + anti-CC1/CC5 **p* = 0.0071, +R28 + anti-CC1/CC5 vs. +R28 + IgG1 *p* = 0.043). **b**, **c** Wound healing of ME-180 cell monolayers upon challenge with isogenic *S. pyogenes* AL368 strains. **b** Shows representative light microscopy images of ME-180 epithelia at ×10 magnification, in which the front of each scratch at 0 h is marked by a complete line and at 24 or 48 h

by a dashed line. **c** shows the mean ± s.d. of at least *n* = 7 independent experiments. Statistical significance tested by one-way ANOVA with Tukey's *post-hoc* test (24 h, Buffer vs. +R28 **p* = 0.004, +R28 vs. −R28 ***p* = 0.0002; 48 h, Buffer vs. +R28 **p* = 0.0016, +R28 vs. −R28 **p* = 0.0099). **d** Treatment of ME-180 cells with purified R28 protein suppresses wound repair (Mean ± s.d. of *n* = 8 independent experiments). Statistical significance tested by one-way ANOVA with Tukey's *post-hoc* test (Buffer vs. R28 **p* = 0.0012, R28 vs. Rib **p* = 0.0056). **e** The impact of rR28-IgI3 protein treatment of ME-180 cell wound repair (mean ± s.d. of *n* = 8 independent experiments). Data generated using wildtype or mutant R28-IgI3 domains. Statistical significance tested by one-way ANOVA with Tukey's *post-hoc* test (Buffer vs. WT *p* = 0.0102, WT vs. Y61A **p* = 0.009).

a more physiologically relevant environment, we utilized human ecto-cervical tissue explants[34], which represent a valuable model to study immune responses of cervix-resident immune cells to microbes. The human ecto-cervical tissue explants were infected with *S. pyogenes* wildtype or isogenic R28-negative strains for 24 h and then cultured in the presence of antibiotics to kill extracellular bacteria for 24 h (Fig. 5a). The ex vivo immune response to *S. pyogenes* in this system was donor specific (Supplementary Fig. 10a), but expression of R28 by *S. pyogenes* caused significant reductions in the production of a subset of cytokines and chemokines including IL-1β, MIP-1β, GM-CSF, SDF-1β, and IL-12 (Fig. 5b). Of note, no toxicity was observed in *S. pyogenes* infected tissues (Supplementary Fig. 10b). Taken together, these experiments indicate that R28-expression by *S. pyogenes* is associated with an impaired innate immune response.

Neutrophils are innate immune cells that provide the first line of defense against invading bacteria in the blood, by phagocytosis and/or

release of antimicrobial factors. However, *S. pyogenes* is equipped with several surface-localized virulence factors, such as M proteins[35,36], that suppress neutrophil-mediated phagocytosis and killing. Because CEA-CAM1 is expressed by human neutrophils upon activation[18], we investigated whether R28 could provide yet another mechanism for *S. pyogenes* to overcome human defenses by enhancing resistance to neutrophil killing. Purified R28 bound to the surface of activated human neutrophils, but not to resting neutrophils (Fig. 5c). Of note, R28 does not interact with the other CEACAM receptors expressed by neutrophils (Fig. 2c, d, e), including the phagocytic CEACAM3 receptor[37,38]. This suggests that R28 expressed on *S. pyogenes* will bind to activated neutrophils exclusively via CEACAM1. Next, we assessed whether R28-CEACAM1 interactions promoted *S. pyogenes* resistance to neutrophil killing by quantifying *S. pyogenes* survival after 1 and 2 h of incubation with human neutrophils. Indeed, R28-expression enhanced the resistance of *S. pyogenes* to neutrophil killing (Fig. 5d),

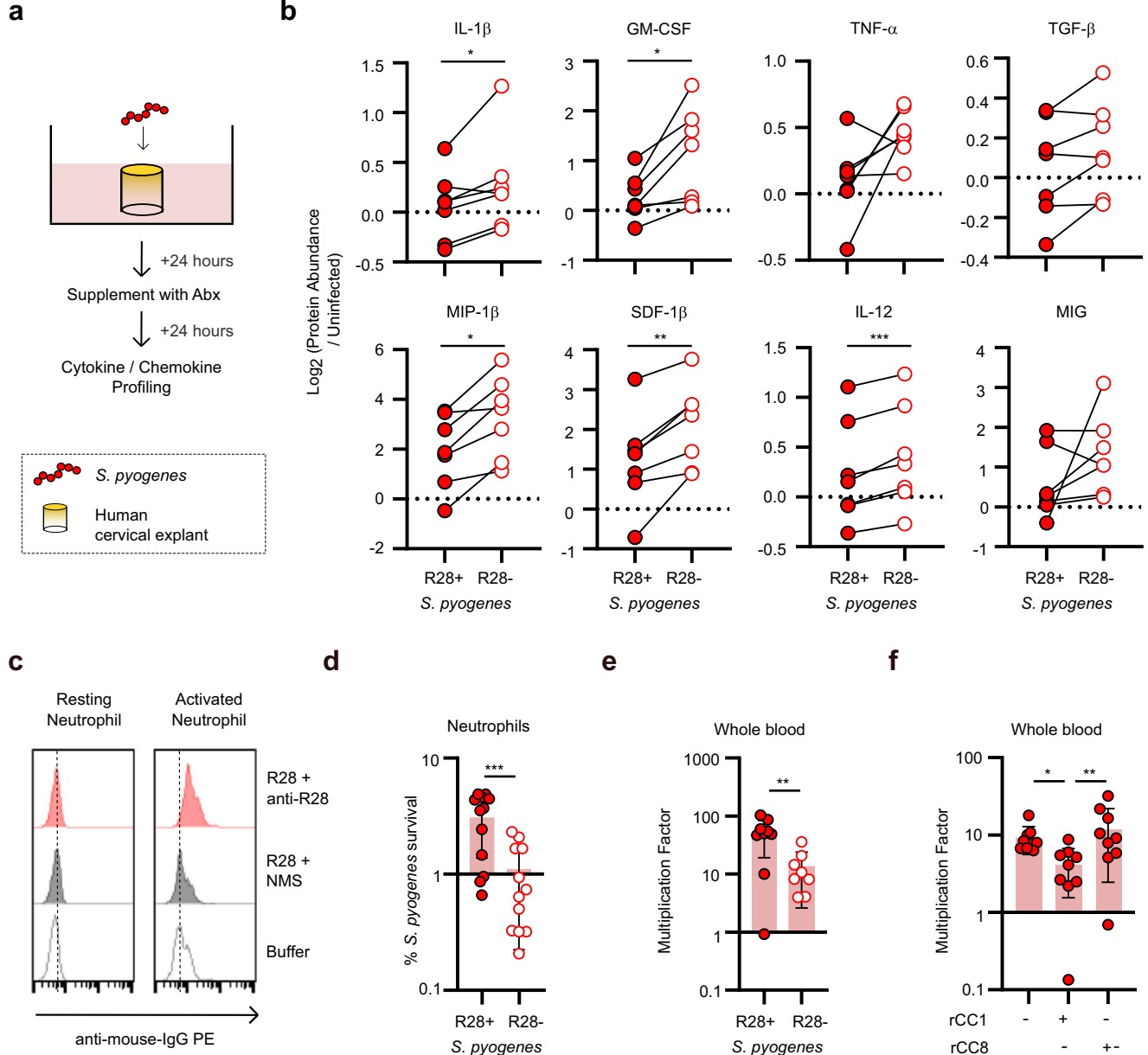

**Fig. 5 | *S. pyogenes* impairs the human immune response through R28-CEACAM1 interaction. a** Schematic representation of measurement of the cytokine and chemokine response of human cervical explants upon challenge with *S. pyogenes* strains or control buffer. **b** Quantification of the protein level production of cytokines and chemokines by wounded human cervical explants in response to infection with *S. pyogenes* strains. Samples were normalized against uninfected controls. Mean and s.d. from $n = 7$ independent human ecto-cervical explant donors. Statistical significance tested by two-tailed paired Student's *t* test (IL-1β *$p = 0.0334$, GM-CSF *$p = 0.0155$, MIP-1β *$p = 0.0139$, SDF-1 **$p = 0.0075$, IL-12 ***$p = 0.0006$). **c** Activated human neutrophils bind purified R28. Data is representative of $n = 3$ independent replicates. **d** Survival of *S. pyogenes* after 2 h of incubation with human neutrophils at a multiplicity of infection (MOI) of 10 was quantified as the percentage of inoculum. Data are represented as mean ± s.d. of $n = 12$ data points from $n = 6$ independent donors. Statistical significance tested by two-tailed paired Student's *t* test (R28 + vs. R28- ***$p = 0.0003$). **e** Replication of isogenic *S. pyogenes* AL368 strains in human whole blood was quantified after 3 h as the percentage of inoculum (mean ± s.d. from $n = 10$ independent experiments). Statistical significance tested by two-tailed unpaired *t* test (R28 + vs. R28- **$p = 0.0063$). **f** Pre-incubation of *S. pyogenes* with rCC1 suppresses the replication of *S. pyogenes* in human whole blood (Mean ± s.d. of $n = 9$ independent experiments). Statistical significance tested by two-tailed paired Friedman test with Dunn's *post-hoc* test (Buffer vs. rCC1 *$p = 0.0286$, rCC1 vs. rCC8 **$p = 0.0065$).

consistent with previous findings[25]. These observations were echoed in whole blood killing experiments (Fig. 5e and Supplementary Fig. 10c), where R28-expression enhanced the resistance of *S. pyogenes* to killing. Moreover, the increased survival of R28-expressing *S. pyogenes* in whole blood was dependent on ability to interact with CEACAM1 (Fig. 5f and Supplementary Fig. 10d), as there was reduced survival when the bacteria were pre-incubated with recombinant CEACAM1. Collectively, these data indicate that the resistance of *S. pyogenes* to innate immune cell killing is enhanced through binding of R28 to CEACAM1.

## Discussion

Puerperal sepsis is a disease with important historical context, as a driver in the development of germ theory and modern infection control[4,5]. We have now clarified the role of the R28 adhesin in the pathogenesis of *S. pyogenes* puerperal sepsis at the molecular and functional level. Our results demonstrate that human CEACAM1 is a highly specific receptor for R28. This single host-pathogen interaction promotes the adhesion of bacteria to epithelial layers, delays epithelial wound healing, subverts the innate immune responses, and enhances

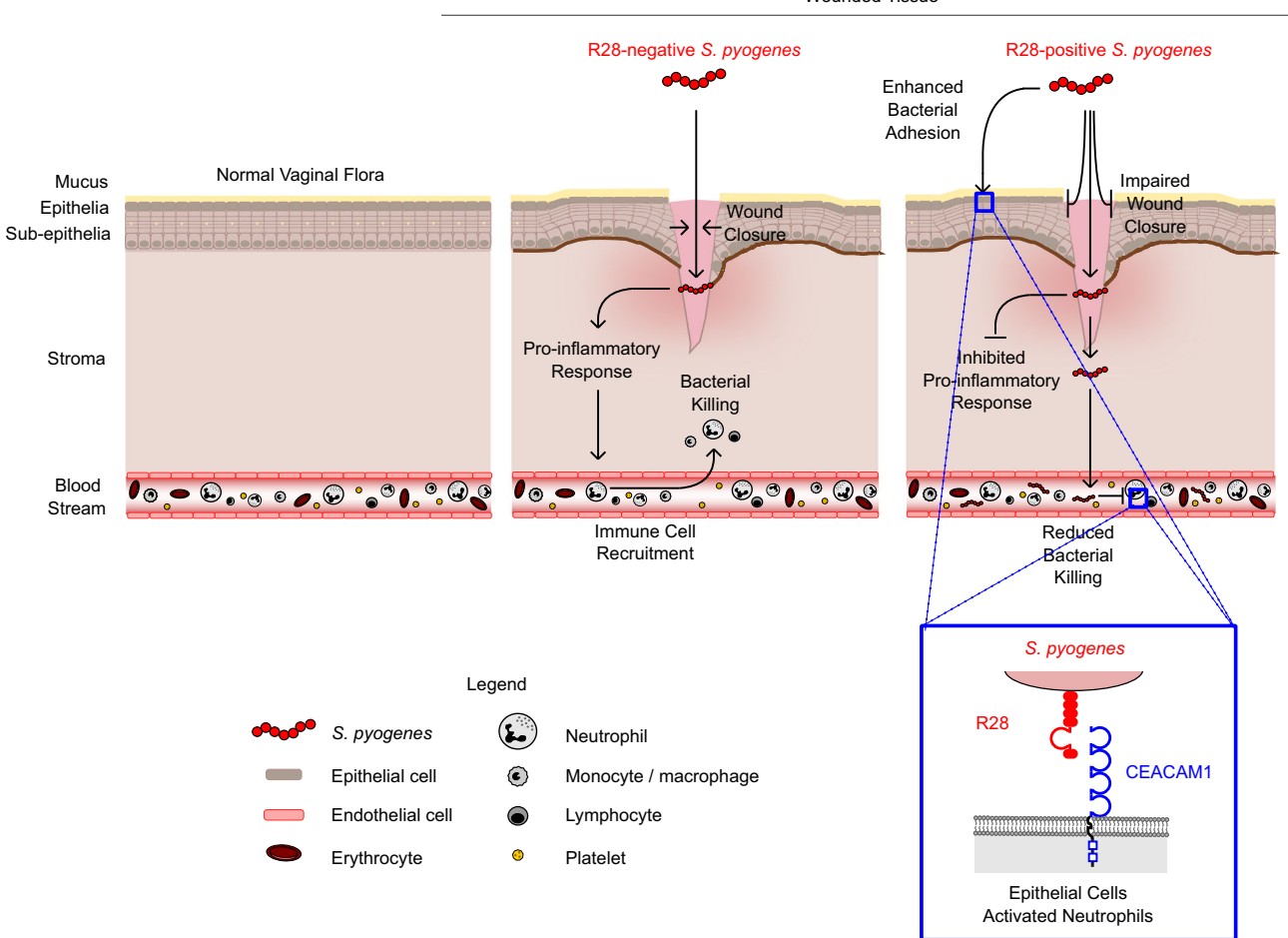

**Fig. 6 | Model for the role of R28-CEACAM1 interaction in *S. pyogenes* puerperal sepsis pathogenesis.** The intact vaginal epithelium (left panel) is rarely colonized by *S. pyogenes*, but wounds introduced during childbirth provide opportunities for R28-expressing *S. pyogenes* to invade the mucosal tissue (right panel). Interaction of R28 with CEACAM1, present on the apical surface of the epithelial cells, impairs wound repair, and suppresses the proinflammatory immune response. This provides enhanced opportunities for *S. pyogenes* to invade the tissue. Moreover, binding of R28 to CEACAM1 on innate immune cells promotes the capacity of *S. pyogenes* to survive in blood, favoring the development of systemic infection. In contrast, an R28-negative mutant does not interact with CEACAM1 and will be less able to evade a proinflammatory response (middle panel).

*S. pyogenes* resistance to immune-mediated killing. Thus, our studies demonstrate that a single bacterial factor can target a single host factor to assist a bacterial pathogen in overcoming multiple host barriers on the pathway to causing an invasive disease. This contrasts the established view that individual molecular host-pathogen interactions mainly drive a bacterial pathogen to overcome a single host defense mechanism. This conclusion emphasizes the importance of considering that individual molecular host-pathogen interactions can have multifaceted functions in bacterial pathogenicity.

The targeting of *S. pyogenes* to CEACAM1 on both epithelial cells and immune cells likely favors bacterial tissue invasion, survival in blood and development of systemic infection, as summarized in Fig. 6. This model is in good agreement with data obtained for other bacterial pathogens that bind CEACAM1 and exploit its role as an immunomodulatory receptor[19,22,39], including *Neisseria gonorrhoeae*, *Neisseria meningitidis*, *Helicobacter pylori*, *Moraxella catarrhalis*, *Fusobacterium nucleatum*, and *Haemophilus influenzae*. While our data indicate that R28 plays a key role in the pathogenesis of puerperal sepsis, additional *S. pyogenes* virulence factors such as M proteins, exotoxins and proteases are probably secondary determinants required to cause this invasive disease[40,41].

The R28 adhesin exclusively binds to CEACAM1. In contrast, other bacterial adhesins bind to a combination of CEACAM1, CEACAM3, CEACAM5 and CEACAM6[17,42–46]. Our finding therefore makes *S. pyogenes* the first pathogen for which CEACAM binding is restricted to CEACAM1. The basis of this selectivity requires further investigation and cannot be explained by targeting of R28 to different CEACAM faces, as R28 and all bacterial ligands characterized to date bind the *A'GFCC'C"* face[17,42–45]. Moreover, the selectivity cannot be entirely explained by differential targeting of residues in the *A'GFCC'C"* face of CEACAM1, as residues F29, Q89, I91 and L95 on CEACAM1 are targeted by R28 and other bacterial adhesins including β protein of *S. agalactiae*, HopQ of *H. pylori* and Opa of *Neisseria*. spp[17,42,44,45]. The unique binding mode of R28 that is highly selective for CEACAM1 requires investigation using formal biophysical assessments.

Bacterial pathogens utilize CEACAM1 to dock onto epithelial cells[47]. For *S. pyogenes*, the binding of R28 to CEACAM1 also provides a mechanism to delay repair of wounds in epithelial layers. These properties of R28 should be of importance in the wounded tissues present in women after delivery, as a delay in wound repair would prolong the time for the pathogen to invade tissues and to cause puerperal sepsis. The properties of R28 also focus interest on the poorly understood mechanisms by which CEACAM1 is known to regulate cell migration and wound repair[31–33]. Of note, the interaction between R28 and CEACAM1 is different from that identified in other CEACAM1-binding bacterial pathogens, which can use CEACAM1 to

trigger endocytosis of bacteria into the epithelial cell where they can replicate and/or transcytoses the epithelial layer[47–49].

*S. pyogenes* has evolved sophisticated mechanisms to evade neutrophil phagocytosis and killing[50]. Activated neutrophils express not only CEACAM1, but also CEACAM3, a specialized receptor employed for phagocytosis and elimination of CEACAM3-binding bacteria[37,38]. Since R28 does not bind CEACAM3, our data imply that CEACAM3 does not affect the role of R28 in the pathogenesis of *S. pyogenes* infections and puerperal sepsis. This lack of interaction with CEACAM3 contrasts with most of the other bacterial ligands of CEA-CAMs, including Opa of *Neisseria spp.*[44,45], HopQ of *H. pylori*[42], and UspA1 of *M. catarrhalis*[43,51]. The high-specificity for CEACAM1, suggests that R28 has evolved to escape CEACAM3-mediated detection by neutrophils, whilst maintaining the capacity to interact with CEACAM1. This could allow *S. pyogenes* to subvert the anti-bacterial responses of neutrophils. A better understanding of the intracellular signaling events affected by R28 engagement of CEACAM1 on neutrophils could help to understand the progression of invasive infections and puerperal sepsis.

Since a recent study reported that human integrins may act as receptors for both the N1 and N2 domains of R28[21], it is possible that integrins serve as co-receptors for R28. However, our in vitro and ex vivo experiments demonstrate that interaction with CEACAM1 is required and sufficient for R28 to suppress wound closure, the innate immune response and neutrophil-mediated killing. We therefore postulate that R28-integrin interactions are of low affinity and dispensable for the functional properties of R28. Similarly, integrin interactions are dispensable for the virulence of *H. pylori*, which targets CEACAM1 through expression of the HopQ adhesin to promote virulence[22,52]. Of note, the identification of human CEACAM1 as the cognate and highly specific receptor for R28 provides critical information that humanized animal models are required to fully elucidate the role of R28 in vivo[53], and that experimental infections in wildtype mice or macaques with R28 strains must be evaluated with caution[9,25,27].

While R28 is found in *S. pyogenes* strains of several *emm* types, viz. *emm*28, *emm*2, *emm*48, and *emm*77, only *emm*28 strains are strongly associated with outbreaks of puerperal sepsis[10–12]. However, *emm*77 strains have been reported to cause at least one outbreak[54]. There are several potential explanations for the association of *emm*28 strains with outbreaks of puerperal sepsis. First, this could be because *emm*28 strains are more common than *emm*2, *emm*77, and *emm*48 strains, as suggested by recent data on *emm* type distributions[13,55]. Second, this could be because the gene for R28, *spr28*, is present in all *emm*28 strains but not all *emm*2, *emm*77, and *emm*48 strains[13,23], a fact that might be attributed to relatively recent acquisition of the mobile genetic element carrying *spr28* into these lineages[24]. Thirdly, our data indicate that R28 expression is higher in *emm*28 strains compared to *emm*2 and *emm*77 strains. Finally, it could be that *emm*28 strains have enhanced transmissibility or virulence properties. Some or all these factors may contribute to the preponderance of *emm*28 among strains causing outbreaks of puerperal sepsis. It is also important to note that sporadic cases of puerperal sepsis are commonly caused by strains of *S. pyogenes* that do not express R28[11]. This may occur if such strains bypass the initial barriers to infection, although they lack R28, in which case the disease may be like that caused by R28-positive *S. pyogenes* strains. However, it seems reasonable to assume that a smaller infection dose is required for R28-positive strains, a factor that may favor the occurrence of outbreaks.

The most appropriate experiment to test the capacity of the R28-CEACAM1 interaction to promote development of puerperal sepsis would be to study disease progress in female mice, including transgenic human *CEACAM1*[+/- or +/+], after intravaginal inoculation with *S. pyogenes* shortly after the birth of pups. However, such a model that follows the natural route of puerperal sepsis does not exist. Given the large sacrifice of pups and the use of females who just had pups, it is highly unlikely that permission to perform such experiments would be provided by an ethical committee. Furthermore, colonization studies in transgenic mice, in the absence of birth-induced damage, are poorly justified as *S. pyogenes* rarely colonizes the reproductive tract of women, even in late pregnancy[56]. These arguments highlight the need to develop new models, that must be humanized given the human-specificity of the R28-CEACAM1 interaction, to investigate puerperal sepsis.

In summary, our results provide strong evidence that the R28-CEACAM1 interaction plays a major role in outbreaks of puerperal sepsis caused by *S. pyogenes*. Thus, our data expand the understanding of one of the most important nosocomial infections in human history. It is noteworthy that life-threatening invasive *S. pyogenes* infections other than postpartum infections have increased in incidence during recent decades and often are caused by R28-expressing strains[13]. Because R28 elicits protective immunity[9], it may therefore become of interest to include it in a future *S. pyogenes* vaccine[57]. This aspect is particularly relevant because R28 is encoded on a mobile genetic element[23,24], raising concerns about dissemination of the gene to new *S. pyogenes* lineages or other bacterial species. Finally, our work highlights that the combination of epidemiology with functional analysis of disease-associated virulence factors can be critical to unraveling the molecular basis of major bacterial infections.

## Methods

### Ethics
Human blood was obtained from healthy donors, approved by the Regional Ethics Committee and Imperial College Healthcare NHS Trust Tissue Bank (Regional Ethics Committee approval no. 17/WA/0161, Imperial College Healthcare Tissue Bank Human Tissue Authority license no. 12275, and Imperial College Research Ethics Committee no. 19IC5166). All samples were collected after receiving signed informed consent from all participants. Human female reproductive tract organ specimens (tissues of vagina, ectocervix, endocervix, uterus, Fallopian tube, and ovary) for immunohistochemistry were obtained from patients treated at the Department of Gynecology of Sumy Regional Oncology Center, Sumy, Ukraine. All tissues were collected after receiving signed informed consent from all patients. The Bioethics Commission of the Medical Institute of Sumy State University approved the experimental protocol (no. 36 from 14.05.2018). Surgically resected specimens of human ecto-cervical tissue were obtained from the York Tissue Bank, University of York, UK approved by the Yorkshire & The Humber-Leeds East Research Ethics Committee (NHS REC 20/YH/0126) or collected at St. Mary's Hospital, Imperial College Healthcare NHS Trust, London, UK, after receiving signed informed consent from all patients through the Imperial College Healthcare Tissue Bank approved by Research Ethics Committee (IRAS 17/WA/ 0161). All tissues were collected after receiving signed informed consent from all patients.

### Human blood
Neutrophils and peripheral blood mononuclear cells (PBMCs) were isolated from human blood samples by Ficoll/Histopaque centrifugation as and resuspended in RPMI 1640 supplemented with 2% FCS, as previously described[58].

### Human vaginal tissue explants
Resected ecto-cervical tissue was cut into 2–3 mm³ explants comprising both epithelial and muscularis mucosae as described previously[59]. Non-polarized explants were maintained with complete DMEM containing 10% fetal calf serum, 2 mm L-glutamine in the presence or absence of antibiotics (100 U of penicillin/ml, 100 µg of streptomycin/ml, and 80 µg of gentamicin/ml) at 37 °C in an atmosphere containing 5% $CO_2$.

## Bacterial strains and culture

Microorganisms used in this study are listed in Supplementary Data 3. Bacteria were cultured in Todd-Hewitt Broth (TH) for *S. agalactiae*, TH Broth + 1% yeast (TH-Y) for *S. pyogenes*, or Lysogeny Broth (LB) for *E. coli*. The medium was supplemented with antibiotics when required.

*PCR analysis:* To test *S. pyogenes* strains for carriage of *spr28*, a PCR analysis was performed using primers that target the IgI3 domain F: 5′-ACAGCTCCAACATTAACCGTC-3′ and R: 5′-TTTTGGTTCGTTGCTA TCCTT-3′, with an expected amplicon of 336 bp. PCR was performed with Phusion Plus DNA polymerase following standard reaction conditions. Thermocycling was performed as 1 cycle of 98 °C for 2 min, 35 cycles of 98 °C for 15 s, 60 °C for 20 s, and 72 °C for 20 s, and 1 cycle of 72 °C for 7 min. Amplification of a product was analyzed using a 1.25% agarose gel.

## Cell lines and culture conditions

HeLa (ATCC) and Chinese Hamster Ovary (CHO; ATCC) cell lines expressing human or macaque CEACAMs were cultured in RPM1640 + 10% FCS, penicillin-streptomycin at 37 °C with 5% $CO_2$[26,60,61]. ME-180 cells (AMS Biotechnology) were cultured in McCoy's 5A media supplemented with 10% FCS, penicillin-streptomycin and amphotericin B at 37 °C with 5% $CO_2$. To measure CEACAM expression on ME-180 cells by flow cytometry analysis, confluent adherent ME-180 cells were detached using 0.25% trypsin, washed, incubated with anti-CEACAM mAb (CC1/3/5-Sab; LeukoCom), monospecific anti-CEACAM1 (clone B3-17; LeukoCom), monospecific anti-CEACAM5 (5C8C4; LeukoCom), monospecific anti-CEACAM6 (1H7-4B; LeukoCom) or isotype IgG1 control mAb (MAB002; R&D Biosystems) for 30 min, washed, incubated with secondary PE-conjugated goat anti-mouse-IgG mAb (12-4010-87; Invitrogen), washed and fixed with 1% PFA. The fluorescence of cells was measured by flow cytometry. Gating strategy is shown in Supplementary Fig. 11a.

## Expression and purification of bacterial proteins

The R28 (Spy1336), β, α, and Rib proteins were purified from *S. pyogenes* and *S. agalactiae* bacteria as previously described[9,62,63]. Recombinant M28 was purified essentially as described for other M proteins[64]. Vectors encoding mutant forms of R28-IgI3 were generated by site-directed mutagenesis of our previously constructed pRSET-C-R28-IgI3 vector (Supplementary Data 4)[17]. Wildtype and mutant forms of the IgI3 domain of R28 (R28-IgI3) were expressed in *E. coli* Rosetta Gami for 4 h at 37 °C following the addition of 1 mM IPTG, and purified using a Nickel column (GE Healthcare Life Sciences) and affinity chromatography (ÄKTA Pure, GE Healthcare Life Sciences), as previously described[17].

## Expression and purification of rCEACAMs

Expression vectors used in this study are listed in Supplementary Data 4. Vectors encoding mutant forms of CEACAM1-N were generated by site-directed mutagenesis of our previously constructed pRSET-C-CEACAM1-N vector, as previously described[17]. Wildtype and mutated forms of rCEACAM1-N were expressed in *E. coli* Rosetta Gami for 4 h at 37 °C following the addition of 1 mM IPTG, and purified using a Nickel column (GE Healthcare Life Sciences) and affinity chromatography (ÄKTA Pure, GE Healthcare Life Sciences), as previously described[17]. Tag-less rCEACAM1-N for use in isothermal titration calorimetry (ITC) and crystallization were expressed using pET21d vectors in *E. coli* and purified, as previously described[17]. Human rCEACAMs with His-tag were expressed in Expi293F cells (Life Technologies) cultured in Expi293 Expression Medium (Life Technologies) and purified by affinity chromatography (ÄKTA Pure, GE Healthcare Life Sciences) using a Nickel column (GE Healthcare Life Sciences), as previously described[17]. Mouse rCEACAM1-HIS was purchased from R&D systems. Human, macaque and rat CEACAM1-Fc were expressed as previously described[22].

## ELISA of R28 and CEACAM interactions

Purified R28 proteins were diluted to 30 μg/ml in carbonate buffer (35 mM $Na_2CO_3$ 15 mM $NaHCO_3$ pH9.6) and immobilized onto the surface of 96-well plates at 4 °C overnight. Wells were washed, blocked with PBS + 0.5% BSA, washed, probed with 10 μg/ml rCEACAM1-HIS (37 °C, 2 h), washed, probed with rabbit anti-CEACAM polyclonal IgG (37 °C, 2 h), washed, probed with horse radish peroxidase (HRP)-conjugated goat anti-rabbit-IgG (A16096; Life Technologies), washed and developed using 50 μl of 1x TMB substrate solution and 50 μL Stop Solution for TMB substrate (Invitrogen).

## Western blot analysis of R28 and CEACAM interactions

Nitrocellulose membranes were blocked with 5% milk solution overnight at 4 °C and probed with 10 μg/ml rCEACAM1-His. Binding of rCEACAM1-HIS to the nitrocellulose membrane was detected using horse radish peroxidase (HRP)-conjugated mouse anti-HIS-IgG (HIS.H8; Invitrogen) and developed using Pierce ECL Western Blot Substrate Reagent (Thermo Fisher Scientific).

## Analysis of proteins binding to dynabeads

C-terminal biotin tagged proteins were attached to streptavidin-coated dynabeads (Dynabead M-280 Streptavidin, Invitrogen; DB) following standard procedures. In all, $3 \times 10^5$ DB was incubated in the presence of 9 μl of rCEACAM1-N-HIS (concentrations range 10 μg/ml), washed with PBS + 0.5% BSA, and probed with FITC-conjugated mouse anti-HIS-IgG (MA181891; ThermoFisher). After washing in PBS + 0.5% BSA, fluorescence of DB was measured by flow cytometry.

## Isothermal titration calorimetry analysis

ITC measurements were performed on an iTC200 instrument (GE Healthcare) in duplicate. 500 μm of rCEACAM-N domains were loaded into the syringe of the calorimeter and 50 μm rR28-IgI3 was loaded into the syringe. All measurements were performed at a stirring speed of 750 rpm, in 30 mm Tris-HCl, 150 mm NaCl, pH 7.5 at 25 °C. Origin 7.0 software was used to analyze all data.

## Measurement of R28 expression by *S. pyogenes*

Mouse antiserum raised against purified R28 protein was previously described[9]. In all, $6 \times 10^6$ of mid-logarithmic phase bacteria were incubated for 1 h at 4 °C with heat-inactivated 0.1% mouse anti-R28 serum or normal mouse serum. After washing, IgG deposition was measured using PE-conjugated goat anti-rabbit-IgG (12-4010-87; Life Technologies) at 4 °C for 1 h and flow cytometric analysis. Gating strategy is shown in Supplementary Fig. 11b. R28 expression by *S. pyogenes* was quantified as IgG deposition after incubation in mouse anti-R28 serum relative to normal mouse serum. The *S. pyogenes* M1 5448 strain that does not express R28 was included as a negative control.

## Analysis of rCEACAM binding to *S. pyogenes*

$6 \times 10^6$ of mid-logarithmic phase bacteria were incubated with rCEA-CAM at 4 °C for 1 h and washed in PBS + 0.5% bovine serum albumin (BSA). Binding of rCEACAM on the bacterial surface was detected by incubation with FITC-conjugated anti-HIS mAb (MA181891; Thermo-Fisher) at 4 °C for 1 h and flow cytometric analysis of fixed bacteria (1% PFA). CC1 binding was quantified relative to an unstained control. The *S. pyogenes* M1 5448 strain that does not express R28 was included as a negative control. The *S. agalactiae* strain expressing the CEACAM1-binding β protein was included as a positive control.

## Screening of receptors for R28

Receptors for R28 were identified using cell microarray technology at Retrogenix, UK. For primary screening, 3559 expression vectors covering more than 2625 different human plasma membrane proteins (Supplementary Table 1) were arrayed in duplicate across 10

microarray slides. Subsequently, human HEK293 cells were grown on top of the cDNA clones and were reverse transfected. An expression vector (pIRES-hEGFR-IRES-ZsGreen1) was spotted in quadruplicate on every slide, and was used to ensure that a minimal threshold of transfection efficiency had been achieved or exceeded on every slide. The cell microarrays were fixed onto the slides and tested for binding of 10 μg/ml purified R28 and Rib protein. After washing of the slides, the streptococcal proteins were detected with rabbit antisera (mixture of two rabbit antisera both diluted 1:1600, including rabbit anti-Rib sera) and AlexaFluor647(AF647)-conjugated anti-rabbit IgG (Invitrogen). Two replicates were performed in the primary screen. Fluorescent images of the slides were analyzed and quantitated (for transfection efficiency) using ImageQuant software (GE). A positive protein 'hit' on was defined as a duplicate spot showing a raised signal compared to background levels, which was assessed by visual inspection using the images gridded on the ImageQuant software. To confirm the hits, all vectors encoding the hits identified in the primary screen were arrayed and over-expressed onto new slides. Slides were screened with 10 μg/ml R28 or buffer followed by rabbit anti-Rib sera as described above.

### Immunohistochemistry of female reproductive tract

Serial sections of 4 μm were prepared from paraffin-embedded tissue previously fixed in neutrally buffered formalin and mounted on 3-aminopropyltriethoxysilane-coated slides. After overnight exposition at 50 °C, tissue samples were de-paraffinized and rehydrated in graded alcohol and xylol. Heat-mediated antigen retrieval was performed utilizing a water bath and 0.01 M sodium citrate for 30 min at 97 °C. Endogenous peroxidase activity was blocked by treatment with 3% $H_2O_2$ for 5 min and subsequent washing with PBS. Sections were blocked with 1% BSA/PBS and incubated overnight at 4 °C with 0.1 μg/ml of monospecific anti-CEACAM1 (clone C5-1X/8; LeukoCom), anti-CEACAM5 (clone 3E10-3; LeukoCom) and anti-CEACAM6 (clone 1H7-4B; LeukoCom), or an isotype control. After washing, sections were probed with biotinylated secondary rabbit anti-mouse antibody (Dako) for 1 h at room temperature. Sections were washed and incubated with VECTASTAIN ABC reagent (Vector Laboratories) for 30 min according to the Manufacturers´ protocol. Staining was visualized by diaminobenzidine (DAB) substrate controlling the developing color intensity via light microscopy. DAB-negative structures were identified by additional counterstaining with hematoxylin. Stained sections were mounted and documented by microscopy.

### Microscopy to assess CEACAM expression

ME-180 cells were seeded at a density of $2.5 \times 10^5$ cells/ml in 24-well dishes and grown until confluency. Cells were incubated with 5 μg/ml CC1/3/5-Sab (LeukoCom) or isotype control mAb (MAB002; R&D systems), washed and incubated with anti-mouse-APC conjugated secondary antibody. Cells were then stained for AlexaFluor555(AF555)-conjugated wheat-germ agglutinin (W32464; Life Technologies), fixed with 4% PFA and stained with 1 μg/ml DAPI. Cells were imaged using widefield microscopy (Zeiss Cell Discoverer 7 – ×20 objective).

### Adhesion of *S. pyogenes* to cell lines

Mid-logarithmic phase *S. pyogenes* cultured in TH-Y were washed and resuspended (CHO and HeLa, in DMEM containing 10% FCS; ME-180 McCoy's 5A containing 10% FCS). Confluent monolayers were washed and infected with the prepared bacteria at a multiplicity of infection (MOI) of 10 or 30. Assays were commenced by centrifugation for 2 min at $100 \times g$ and incubated at 37 °C with 5% $CO_2$ for 30 min (CHO or HeLa cells) or 120 min (ME-180 cells). Monolayers were washed five times with PBS, detached with 0.25% trypsin and lysed with PBS containing 0.025% Triton-X. Serial dilutions of the cell lysates were plated onto TH-Y agar plates. After incubation of TH-Y plates overnight at 37 °C, the number of adherent bacteria were enumerated. CEACAM-specific

interactions were measured in some assays by pre-incubating cells with 5 μg/ml CC1/3/5-Sab (detects N-terminal domain of human CC1, CC3, and CC5) or isotype control mAb for 30 min.

### Binding of FITC-labeled *S. pyogenes* to cell lines

Confluent CHO or ME-180 cells were detached, washed, resuspended to $4 \times 10^6$ cells/ml and incubated with FITC-labeled *S. pyogenes* strains at an MOI of 30 for 30 min at 37 °C. After washing and fixation in 1% PFA, the fluorescence of cells was measured by flow cytometry. Gating strategy is shown in Supplementary Fig. 11a.

### Scratch assay of ME-180 wound closure

In all, $2.5 \times 10^5$ ME-180 cells were cultured on 24-well tissue culture plates until 85% confluency. In each well, a scratch was made with the tip of a yellow pipet tip and monolayers were washed three times to remove detached cells. The cells were incubated at 37 °C with 5% $CO_2$ for 1 h after addition of 125 μl of McCoy's 5A + 10% FCS to each well. Four photographs of each scratch, representing 0-h timepoint (t0), were collected using a Nikon D5100 Digital Camera. For infection experiments, tissue culture wells were infected with $1 \times 10^8$ mid-logarithmic phase *S. pyogenes* cultured in TH-Y and subsequently resuspended in McCoy's 5A containing 10% FCS. After 24 h incubation at 37 °C with 5% $CO_2$, monolayers were washed three times to remove bacteria and supplemented with McCoy's 5A media containing 10% FCS containing penicillin/streptomycin. Four photographs of each scratch, representing the 24-h timepoint (t1), were collected. The cell monolayers were cultured for a further 24 h at 37 °C with 5% $CO_2$, prior to collection of four photographs of each scratch, representing the 48-h timepoint (t2). The mean distance of each scratch across the four photographs was measured using ImageJ. CEACAM-specific interactions were measured in some assays by incubating cells with 20 μg/ml CC1/3/5-Sab (detects N-terminal domain of human CC1, CC3, and CC5) or isotype control mAb throughout the assay.

For assays employing purified R28 or Rib or rR28-IgI3 variants, wounded ME-180 cells were supplemented with McCoy's 5A media containing 10% FCS containing penicillin/streptomycin and 10 mg/ml of the specific protein. Four photographs of each scratch, representing the 0-h timepoint (t1), were collected at 10x magnification. The wounded ME-180 cells were cultured for 24 h at 37 °C with 5% $CO_2$, prior to collection of four photographs of each scratch, representing the 48-h timepoint (t2).

In both experimental types, the percentage of wound closure was calculated as (scratch distance at t1 or t2 ÷ scratch distance at t0) × 100, where t0 equals 0 h and t1 and t2 equal a 24 or 48 h timepoint.

### Transwell assays

In all, $2 \times 10^4$ cells/well were cultured on Transwell (Greiner Bio-One ThinCert, pore diameter 3 μm) inserts in the apical chamber of a 24-well plate until confluency. PBMCs were seeded at a density of $1.25 \times 10^6$ cells/well in the basal chamber of a 24-well plates. The apical chamber was challenged with γ-irradiated *S. pyogenes* strains at an MOI of 10 and incubated for 72 h at 37 °C with 5% $CO_2$. Supernatants were harvested from both chambers and centrifuged at $8000 \times g$ to remove excess bacteria before storage at -80 °C for Luminex analysis.

### Human ecto-cervical explant experiments

The ecto-cervical explants were incubated for 3 h in complete DMEM with antibiotics. After four washes in PBS, the explants were transferred to fresh plates and challenged with *S. pyogenes* in complete DMEM with no antibiotics for 24 h. Explants were then transferred to new plates and cultured for 5 days in complete DMEM with antibiotics. Culture supernatants were harvested at days 2 and 5 post-challenge for proteomic evaluation. Viability of tissue explants following exposure to *S. pyogenes* was determined by measuring tetrazolium salt

[3-(4,5-dimethyl-2-thiazolyl)-2,5-diphenyl-2H-tetrazolium bromide (MTT)] cleavage into a blue product (formazan) by viable cells[65], as described previously[66]. Optical density values obtained with a Synergy-HT (BioTek, Winooski, VT) plate reader were corrected for explant dry weight. Untreated tissue was considered as positive control of viability.

## Measurement of cytokine and chemokine profiles

A total of eighteen soluble immune proteins were quantified in three panels by in house multiplex bead immunoassay using a Luminex 200 System (Bio-Rad, Hercules, CA) as described previously[67]. Cytokine/chemokine levels were normalized against total protein content as measured by a BCA protein assay (Bio-Rad, Hercules, CA). Cytokine concentrations were calculated from sigmoid curve-fits (Prism, GraphPad). All data presented fulfil the criterion of $R2 > 0.7$. For trans-well experiments employing ME-180 cells and primary human PBMCs, ratios between cytokine/chemokine concentrations in explant cultures dosed with *S. pyogenes* vs. untreated tissues were established and statistically compared using unpaired t test and considered significant at $p < 0.05$. For human ecto-cervical explant experiments, ratios between cytokine/chemokine concentrations in explant cultures dosed with *S. pyogenes* vs. untreated tissues were established and statistically compared using unpaired *t* test and considered significant at $p < 0.05$.

## Neutrophil and whole blood assays

Binding of R28 to neutrophils was assessed by flow cytometric analysis. 45 µl of $3 \times 10^7$ cells/ml were incubated with 5 µl of 10 µg/ml R28 or buffer on ice for 45 min. After washing, neutrophils were incubated with 25 µl of 0.1% mouse anti-R28 sera or normal mouse sera for 45 min on ice. After washing, neutrophils were incubated with 25 µl of PE-conjugated goat anti-mouse-IgG for 45 min on ice. Fluorescence of neutrophils was measured by flow cytometry. Gating strategy is shown in Supplementary Fig. 11c. *S. pyogenes* strains were sub-cultured to early-log phase and resuspended in PBS + 0.5% BSA. Purified neutrophils were resuspended to $1 \times 10^7$ cells/ml and 180 µl was inoculated with 20 µl *S. pyogenes* at an MOI of 10, and incubated at 37 °C with gentle rotation. After 2 h, 50 µl of sample was lysed with 0.3% saponin on ice. Lysed samples were serially diluted in PBS and plated onto TH-Y agar for CFU enumeration. For whole blood killing assays, 270 µl of human blood, freshly drawn into tubes with heparin, was inoculated with 15 µl of *S. pyogenes* strains (CFU = 3000) and incubated at 37 °C for 3 h. Inoculated blood was serially diluted in PBS and plated onto TH-Y agar plates for CFU enumeration.

## Crystallization of R28-IgI3 and CEACAM1-N complex

R28-IgI3 was treated with carboxypeptidase A (1:100 w/w) for 24 h at room temperature to remove the C-terminal His-tag and part of the C-terminal region of R28-Ig3. Reaction was quenched with 5 mm EDTA before repurification on a Superdex 200 column (GE Healthcare) equilibrated with 20 mm Tris, 150 mm NaCl, pH 7.5. Purified CEACAM1-N was added in a slight excess (1:1.1) with R28-Ig3, incubated on ice for 30 min before purification on a Superdex 200 column with the same buffer described above and concentrating to 5 mg/ml by centricon. The complex was screened against several different commercial crystallization screens. Crystals were identified in the Wizard Screen (1.8 m Ammonium Sulfate, 0.1 m Sodium Citrate pH 5.5). Crystals were optimized with the best crystals produced in 1.45 M Ammonium Sulfate, 0.1 M Na/K Phosphate pH 5.5. Crystals were cryoprotected in 30% v/v glycerol, and a dataset was collected beamline 9-2 at the Stanford Synchrotron Radiation Lightsource. Data was processed and indexed with XDS before scaling and conversion to structural factors using Aimless. MOLREP was used to perform molecular replacement using CEACAM1-N (PDB: 2GK2) and β-IgI3 (PDB: 6V3P) as search models. Refinement was carried out by REFMAC5 and Phenix Refine. Rebuilding was carried out in Coot. Refinement statistics are shown in Supplementary Table 3.

## Bioinformatics analysis

The presence of *spr28* was confirmed in 3886 previously assembled and curated genomes of UK *S. pyogenes* isolates using Seemann T, Abricate (https://github.com/tseemann/abricate) to detect six regions of the *spr28* gene (90-336 bp in length)[68]. A subset of genomes to represent ten isolates per *emm*-type and excluding any with fewer than ten (except *emm*48 which was represented by a single *spr28* positive isolate) but including all 26 *emm*77, were subjected to pangenome analysis and core genome alignment using Panaroo and a phylogenetic tree constructed with RAxML and annotated in iTOL[69–71].

The structure of human CEACAM1-N$^{F29A}$, human CEACAM1-N$^{F29I}$, macaque CEACAM1-N and macaque CEACAM1-N$^{I29F}$ were predicted using the human (PDB: 4QXW) CEACAM1 structures, respectively, as input in SWISS-MODEL[72]. All models passed a QMEAN score of $<-4.00$. The docking of the R28-IgI3 structure to CEACAM1-N structure or predicted structures was simulated using HawkDock server[73]. The binding energy of simulated complexes (kcal/mol) was compared by One-way ANOVA with Tukey's multiple comparison test.

## Quantification and statistical analysis

All quantitative data were analyzed and graphed using GraphPad Prism 8.4.3 software. All data are represented as mean ± s.d. calculated using the GraphPad Prism 8.4.3 software, unless indicated otherwise. Statistical details of the experiments are provided in the respective figure legends and in each methods section pertaining to the specific technique applied.

## Reporting summary

Further information on research design is available in the Nature Portfolio Reporting Summary linked to this article.

## Data availability

The co-crystal (R28-IgI3 in complex with CEACAM1-N) structural data generated in this study have been deposited in the Protein Data Bank (PDB) database under accession code 8CXJ. Source data are provided with this manuscript. Source data are provided with this paper.

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

## Acknowledgements

This manuscript is dedicated to the life of Bernhard B. Singer who passed away in January 2023 after the experimental part of this work and drafting of the manuscript was completed. We would like to thank the CDC (USA) and Shiranee Sriskandan (Imperial College London) for access to clinical *S. pyogenes* strains. We thank the Department of Surgery and Cancer and the Department of Obstetrics and Gynaecology at St. Mary's Hospital, Imperial College for their assistance in obtaining human tissue. We thank York Tissue Bank, York Biomedical Research Institute, University of York, UK for access to human ecto-cervical explants. We thank staff at the Flow Cytometry Facility and High-Throughput Single Cell Analysis (HTSCA) Facility for technical assistance. We thank Carla J.C. de Haas, Piet Aerts, Kok P.M. van Kessel (UMC Utrecht, Utrecht, The Netherlands), and Birgit Maranca-Hüwel, Bärbel Gobs-Hevelke (University of Duisburg-Essen, Germany) for technical support. We thank the staff of Beamlines 12-2 and ID23-D at the Stanford Synchrotron Radiation Lightsource and the Advanced Photon Source, respectively, for support in data collection. This work was supported by the European Union's Horizon 2020 Research and Innovation Programme under Grant Agreement 700862 (A.J.M. and J.A.G.V.S.), the Medical Research Council through the Doctoral Training Award MR/N014103/ to Imperial College London (E.A.C. and A.J.M.), the Biotechnology and Biological Sciences Research Council award BB/V006495/1 (A.J.M.), Royal Society Research Grant RGS \R1\201044 (A.J.M.), Deutsche Forschungsgemeinschaft DFG grant SI-1558/6-1 (B.B.S.), The Swedish Research Council grant K2011-56X-09490-21-6 (G.L.), a Vidi grant (91713303) from the Dutch Research Council for Health Research and Development (ZonMW; N.M.v.S.), and The Foundation Olle Engkvist Byggmästare (G.L.).

## Author contributions

Conceptualization by B.B.S., N.M.V.S., G.L., and A.J.M. Experimental design, experimentation and data analysis by E.A.C., D.A.B., C.H, M.S.-C., M.L., C.E.T., J.S, B.B.S., N.M.V.S., G.L., and A.J.M. Writing – original draft by E.A.C., D.A.B., C.H., N.M.V.S., G.L., and A.J.M; review and editing manuscript by E.A.C., D.A.B., C.H., C.E.T., J.A.G.V.S., B.B.S., N.M.V.S., G.L., and A.J.M. funding acquisition by J.A.G.V.S., A.J.M., B.B.S., and G.L.

## Competing interests

The authors declare no competing interests.
