## [Peer Review File · Nature Communications]

Human CEACAM1 is targeted by a *Streptococcus pyogenes* adhesin implicated in puerperal sepsis pathogenesisREVIEWER COMMENTS

Reviewer #1 (Remarks to the Author):

Catton et al examine "The R28 adhesin of *Streptococcus pyogenes* promotes puerperal sepsis by targeting the CEACAM1 receptor" (the title of the paper). Prior work shows R28 can contribute to cell binding, including the ME180 cell line used in this study. This has previously been reported by the authors and others to be at least in part due to the binding of integrins and other surface proteins. These targets are not refuted in this study, but are implied to be inconsequential. Additional prior work by the authors show that R28 from *S. pyogenes* bind CEACAM1 (ref 15). Thus, the primary findings of this study are 1) additional confirmations that R28 binds CEACAM1 and 2) a cocrystal structure of the proteins. These data are reasonably convincing. The biological experiments are mostly well-conducted, but do not bring new insights into this mechanism of pathogenesis. The structure could provide insights into this mechanism, but this is underutilized in its placement at the end of the paper, and none of the key observations that could be made from this are explored. The title promise that "The R28 adhesin of *Streptococcus pyogenes* promotes puerperal sepsis" is never shown, and the role of this mechanism in puerperal sepsis really should be. Thus, additional work is needed to support the conclusions and claims of this work.

Major

Fig 5. It will be of interest to the field that the form of R28 protein used does not bind the recombinant form of truncated CEACAM1 of other species as well. However, this reviewer feels greater support should be given when using this a limited observation to question previous models and when suggesting that this is important for virulence. This can be easily shown, and should be. Human CEACAM1 transgenic mice have existed for at least 10 years and are widely available. Relevant infection models for GAS also exist. The authors should do experimental infection to show that R28 binding CEACAM1 is important for virulence.

Fig 6. Can the authors clarify this model to take into account the bulk of their findings? It appears simplified beyond usefulness and based only a few observations. For example, does R28 positively impact on the level of bacterial burden? Adherence to the cervix? Can bacterial number impact inflammation and wound closure? Are there differences in puerperal fever presentation between emm28-infected individuals and those with infections lacking R28? This model says that there are, and they should be clinically obvious and easy to examine.

Line 66: The provide references (9-14) do not support the statement that "an antigen designated R28.. is associated with puerperal sepsis". Some support that certain serotypes are associated with puerperal sepsis, some which can encode R28, but this connection is not as strongly established as stated. Why is there is no association of emm2, emm48, and emm77 with puerperal sepsis, which the authors show in Fig 1e to carry R28? Lower expression (1F) does not appear likely, as there is no binding differences (1G). The importance of this specific mechanism in the development of disease may thus be somewhat oversold throughout.

The images of Fig 3C are of poor quality, and increased resolution and contrast are needed to visualize. Scale bars are absent. The procedure for this experiment is also very unclear; in the methods, it's stated that the cells are infected with (an unknown) amount of *S. pyogenes* for 24 h. However, no bacteria are apparent – it would be expected for the dish to be overgrown and everything dead at this point, by acidification of the media or more direct toxicity. Can the authors clarify how this was done? The mechanism is stated to be "CEACAM1-dependent" (line 792), but this is not shown. CEACAM1 knockout cells should be tested.

In addition to testing CEACAM1 $-/-$ cells, cells expressing 'nonbinding' CEACAMs of Fig 6 should be tested.

Minor

Line 67-69: It is not clear over these lines how R28 is an important problem to solve for puerperal sepsis caused by other pathogens, which lack this protein

Weckel et al (ref 19) is only cited that CEACAM modulates cellular activities (line 74) and anti-bacterial responses (177). However, this paper shows neither thing; it shows R28 acts by binding multiple integrins.

Line 295: Ref 20 regards HopQ, not R28. I suspect many of the references are just scrambled. The authors should confirm and correct these as appropriate. It's not possible to readily evaluate the overall scholarship as it is.

Fig 2D: it is not clear in the legend what a "panel of strains" could be referring to. Is this just wt and a mutant? Do emm2, emm48, emm77 show a similar pattern? Naturally R28-deficient strains would be appropriate negative controls.

Fig 3B is somewhat unusual. The performance of the experiment is the same as 3A, but changed to a distorted linear scale. No statistical methods are provided here (or anywhere in the manuscript), but presumably these are also variable, or potentially incorrect. Can the authors confirm the proper presentation and statistical analysis for this distribution of data? Including the R28-mutant in 3B is an important control to show specificity.

Line 294, reference needed

The ITC of Sup Fig 2 does not clearly show the affinity for rCC1-N, unless that is the same as previously published, and the only new data are all the non-binders?

Reviewer #2 (Remarks to the Author):

In the manuscript, "The R28 adhesin of *Streptococcus pyogenes* promotes puerperal sepsis by targeting the CEACAM1 receptor", Catton et al identify a novel CEACAM-binding adhesion from *Streptococcus pyogenes*, which not only seems to be the major (necessary and sufficient) adhesion driving pathogenesis of bacterial puerperal sepsis, but also interferes with epithelial wound healing, and contributes to subverting the innate immune response.

The manuscript is in general very well written, and the set up of experiments and their descriptions are adequate and can be easily followed. I have only a few comments for improvement, in general the manuscript requires only minor revision.

Minor points:

The effect on wound healing is very interesting. Is this mechanism specific for the R28 protein, or also observed with other bacteria-derived CEACAM1 ligands, such as HopQ from *H. pylori*? Could this be included as a control?

The authors show that R28-IgI3 docks over the CEACAM cis/trans dimerization pocket, and some of the residues within the CEACAM-N domain were also shown to be involved in binding of other bacterial ligands, such as HopQ. In this context:

Is CEACAM1 glycosylation involved in this interaction? While most bacterial adhesins binding to CEACAMs interact with sugar residues, HopQ is the only bacterial adhesion yet that was shown to bind via protein-protein interaction. Was this explored for R28?

R28 only binds to CEACAM1, while other bacterial adhesins bind to CEACAM 5 and 6 also. I am missing an interpretation of the binding interface and residues involved which helps to explain this differential behavior. Also, it would be interesting to model the interaction of the R28 protein with CEACAM1 in direct comparison to the interaction of HopQ and CEACAM1, which might help to understand the partially overlapping but seemingly also differential binding modes.

It was described that interaction of HopQ with CEACAM1 alters CEACAM cis/trans dimerization. Does R28 binding interfere with CEACAM cis/trans dimerization? Does it affect CEACAM1 dependent signaling?

Discussion:

The discussion is very short. Immune regulating effects of other CEACAM binding pathogens should be compared. CEACAM1 selectivity is not discussed. Binding interface of R28 in comparison to other adhesins should be discussed.

Reviewer #3 (Remarks to the Author):

Puerperal sepsis is a devastating disease affecting many women around the world. The causative agent of puerperal sepsis is *Streptococcus pyogenes* (GAS), and the molecular mechanisms by which GAS is adapted to and able to cause puerperal sepsis is unknown.

Catton and Bonsor, et al. investigate the bacterial adhesin, R28, and its contribution to disease. The authors identified that R28 specifically binds to human CEACAM1, which is widely expressed in epithelial and immune cells. Through structural elucidation, the authors identify key residues in the binding pocket between R28 and CEACAM1, use blocking assays to prevent GAS binding to CEACAM1 and show decreased adhesion to epithelial cells. The authors identify R28-CEACAM1 interaction is required for suppressed immune response and decreased wound repair in vitro. These results are noteworthy and add mechanistic knowledge to the area of puerperal sepsis.

Major comments:

Statistical analyses were performed throughout the manuscript, yet the description of statistical analyses that were performed are not mentioned in figure legends or materials and methods.

Figure 1: The binding of R28 expressing strains to CC1 (Fig 1G) has a broad distribution for each emm 28, emm2 and emm77 strains. I am curious regarding the R28 adhesion in the low binding strains: is the R28 sequence identical? Are there mutations present in the IgI3 like domain that could explain the decreased binding affinity?

Figure 3: Fig 3A, a subset of the R28- replicates adhere just as well to the ME-180 cells as WT. Why do the authors think this is? Is there a second bacterial adhesin that could supplement the binding to ME-180 cells?

The authors state that when GAS R28+ is present in the scratch assay that there is decrease wound healing. Is there host factors, such as EGF or associated proliferation pathways that the authors could use qPCR or Western blot to quantify the difference to further back up the claim of CEACAM1 binding is the only factor preventing wound healing?

Regarding immune suppression in human explants, would a similar response be observed if purified R28 was added to the explants or is the whole bacterial interaction with CEACAM1 required? A similar approach would be to incubate the irradiated bacteria with purified CEACAM1 and does the similar suppression occur?

Minor comments

It is unclear which data set is displayed in Figure 4D. Is the neutrophil survival displayed occurring at 1 or 2 hrs? It would be beneficial to clarify if the data is a combination of both time points or a single time point is displayed.

Figure 4A model of cervical explant assay says cytokines were monitored at 24hrs after removal of GAS, which does not align with the material and methods section which mentions levels were monitored at 2 and 5 days. I find it curious that cytokine levels are monitored after removal of GAS, would it not be more representative of an infection for GAS to be present the whole time during the assay?

An interesting idea: are the authors considering developing a transgenic mouse line expressing human CC1 or a humanized mouse CC1? The alteration to the Macaques CC1 shows promise that GAS R28 could bind other species CC1 if the protein was "humanized".

Reviewer #4 (Remarks to the Author):

This manuscript is a thorough and highly informative and impactful identification of a bacterial pathogen - host interaction. It was easy to follow and read, integrated many approaches to address the biological questions, and was of high quality in terms of data and presentation. I have a few comments and questions that would help a reader fully understand the results.

1. With respect to the neutrophil experiments, I am not sure what the authors think is going on. Are the bacteria only adhering to neutrophils and not being phagocytosed and that is why they aren't being killed? I believe both ceacam 1 and ceacam 3 binding can lead to phagocytosis. This is already a massive amount of data but this result does seem to need a bit more to be understood.
2. A few sentences comparing the binding site among the ceacams to report the molecular determinants about the ceacam selectivity would be of interest to a general reader. As well as indicating beyond Hop the other bacterial (and viral?) pathogens that engage ceacams. The scientists that work on these systems will find this interaction fascinating.

Minor

I think there is a typo in the caption of Figure 1, I am not sure how immobilized R28-IgI3 would be used in a flow cytometry assay.

POINT-BY-POINT RESPONSE

Reviewer #1

1. Catton et al examine “The R28 adhesin of *Streptococcus pyogenes* promotes puerperal sepsis by targeting the CEACAM1 receptor” (the title of the paper). Prior work shows R28 can contribute to cell binding, including the ME180 cell line used in this study. This has previously been reported by the authors and others to be at least in part due to the binding of integrins and other surface proteins. These targets are not refuted in this study, but are implied to be inconsequential. Additional prior work by the authors show that R28 from *S. pyogenes* bind CEACAM1 (ref 15). Thus, the primary findings of this study are 1) additional confirmations that R28 binds CEACAM1 and 2) a cocrystal structure of the proteins. These data are reasonably convincing. The biological experiments are mostly well-conducted, but do not bring new insights into this mechanism of pathogenesis. The structure could provide insights into this mechanism, but this is underutilized in its placement at the end of the paper, and none of the key observations that could be made from this are explored. The title promise that “The R28 adhesin of *Streptococcus pyogenes* promotes puerperal sepsis” is never shown, and the role of this mechanism in puerperal sepsis really should be. Thus, additional work is needed to support the conclusions and claims of this work.

We thank the reviewer for their evaluation of our manuscript.

As suggested, we have modified the title of the manuscript to “The R28 adhesin of *Streptococcus pyogenes* targets human CEACAM1: implications for the pathogenesis of puerperal sepsis”. We also appreciate the suggestion of the reviewer that the manuscript could be better organised to make use of the structural information when investigating biological properties. Therefore, we have reorganised the manuscript along the themes of a) identification of the interaction, b) specificity of the interaction, c) structural characterisation of the interaction, d) biological role of the interaction at epithelia, e) biological role of the interaction in immunomodulation, f) model. We have incorporated new assays into section (d) based on the structural information generated in (c).

Major

2. Fig 5. It will be of interest to the field that the form of R28 protein used does not bind the recombinant form of truncated CEACAM1 of other species as well. However, this reviewer feels greater support should be given when using this a limited observation to question previous models and when suggesting that this is important for virulence. This can be easily shown, and should be. Human CEACAM1 transgenic mice have existed for at least 10 years and are widely available. Relevant infection models for GAS also exist. The authors should do experimental infection to show that R28 binding CEACAM1 is important for virulence.

We agree with the reviewer that *in vivo* models provide a powerful platform in infection studies, such as for testing the role those specific molecular interactions play in virulence and disease. However, all experimental animal models must “model” the infectious disease of study to provide relevant and translatable scientific data. In this study, the most appropriate experiment would be to assess the progress of puerperal sepsis following the natural route of *S. pyogenes* infection in mice. Such a model of puerperal sepsis does not exist. A new experimental model would require that a female mouse is inoculated intravaginally with *S. pyogenes* shortly after the birth of pups. Considerable resources and time would be required to establish and validate this completely new model of puerperal sepsis. Moreover, each data point collected would lead to the death of the mother and all offspring (typically 6 to 10 pups per female mouse). Given the large sacrifice of animals and the use of females who just had pups, it is highly unlikely that permission to perform such experiments would be provided by an ethical committee. Importantly, in contrast to other pathogens (i.e. *N. meningitidis*, *N. gonorrhoeae*, *E. coli*, *S. agalactiae*), *S. pyogenes* does not commonly colonize the tissue of entry (human vagina, cervix) on the pathway to causing the invasive disease of study (puerperal sepsis). Thus, we believe that colonization studies of the female reproductive tract, in the absence of birth-induced damage, are not justified. We have assessed the functional properties of the R28-CEACAM1 interaction using the most appropriate *in vitro* and *ex vivo* models available and using clinical *S. pyogenes* isolates. In addition, we have now performed our experiments with an alternative isogenic pair of *S. pyogenes* strains (shown in Supp Fig 7D, 8C, 10C). The results generated using the models provide very strong evidence for the role of the R28-CEACAM1 interaction in infection biology. We have included a discussion of this important point.

Page 18 Lines 388-398 “The most appropriate experiment to test the capacity of the R28-CEACAM1 interaction to promote development of puerperal sepsis would be to study disease progress in female mice, including transgenic human *CEACAM1*^{+/-} or ^{+/+}, after intravaginal inoculation with *S. pyogenes* shortly after the birth of pups. However, such a model that follows the natural route of puerperal sepsis does not exist. Given the large sacrifice of pups and the use of females who just had pups, it is highly unlikely that permission to perform such experiments would be provided by an ethical committee. Furthermore, colonization studies in transgenic mice, in the absence of birth-induced damage, are not justified as it is not representative of the fact that *S. pyogenes* only rarely colonizes the reproductive tract of women, even in late pregnancy⁵⁶. These arguments highlight the need to develop new models, that must be humanized given the human-specificity of the R28-CEACAM1 interaction, to investigate puerperal sepsis.”

As for the human-specificity of the R28-CEACAM1 interaction, our data in Fig 3 and Supp Fig 6 conclusively demonstrate that R28 does not interact with mouse, rat or macaque CEACAM1. Our statements do not criticize previous work or their findings but raise the important point that previous experimental data should be re-evaluated given that R28 does not interact with CEACAM1 in murine or macaque systems, thereby precluding a full evaluation of R28 contribution to pathogenesis. Our findings are now summarized as follows:

Page 11 Lines 217-219 “These results show that R28 specifically binds to human CEACAM1 and indicate that results from experimental animal models likely underestimate the role of R28 in *S. pyogenes* virulence”

Page 17 Lines 367-370 “Of not, the identification of human CEACAM1 as the cognate and highly specific receptor for R28 provides critical information that humanized animal models are required to fully elucidate the role of R28 *in vivo*⁵³, and that experimental infections in wildtype mice or macaques with R28 strains must be evaluated with caution^{9,25,27}.”

3. Fig 6. Can the authors clarify this model to take into account the bulk of their findings? It appears simplified beyond usefulness and based only a few observations. For example, does R28 positively impact on the level of bacterial burden? Adherence to the cervix? Can bacterial number impact inflammation and wound closure? Are there differences in puerperal fever presentation between *emm28*-infected individuals and those with infections lacking R28? This model says that there are, and they should be clinically obvious and easy to examine.

Puerperal sepsis is a medical emergency that requires rapid antibiotic treatment. We agree with the reviewer that there are many clinically obvious questions that require attention. However, these questions are extremely difficult to address as clinically relevant specimens, collected through invasive procedures, during such medical emergencies are rarely available. In contrast, clinical *S. pyogenes* isolates can be accessed (including historical isolates from the 1930s) and they have been integrated into our studies. Thus, this manuscript uses clinical *S. pyogenes* isolates and the best experimental models available, in the absence of clinically relevant specimens. The model we have built in Fig 6 is built on the new information generated in this manuscript.

In our study, we have assessed the functional properties of the R28-CEACAM1 interaction, including studies of *S. pyogenes* adhesion to human cervical epithelial cells (Fig 4), cervical epithelial wound repair (Fig 4), immune responses of human cervical tissue to *S. pyogenes* (Fig 5), and the capacity of *S. pyogenes* to resist immune cell killing (Fig 5). The results generated using the experimental approaches conclusively establish the role of the R28-CEACAM1 interaction in infection biology. Moreover, we provide a detailed structural analysis of the interaction between R28 and human CEACAM1. This information provides a significant step forward in our understanding of the role of R28 in driving puerperal sepsis outbreaks. We have built the model (Fig 6) on this information. Whilst we acknowledge that the model provides an incomplete picture of the pathogenesis of puerperal sepsis as a clinical entity, it summarizes several important findings described in our paper, viz., that R28 promotes adhesion to cervical epithelial cells, that R28 limits wound closure, and that R28 interferes with innate immunity, including phagocytosis. Concerning bacterial burden at the site of infection, our demonstration, that R28 enhances bacterial adhesion and interferes with phagocytosis, strongly imply that R28 increases bacterial burden during the early steps of puerperal sepsis. Of note, this model does not exclude that a strain not expressing R28 may cause puerperal sepsis if it has managed to bypass the initial obstacles to infection, and the disease caused in this case need not be different from that caused by an R28-expressing strain, as now mentioned in the Discussion (Lines 382-385). In summary, we respectfully suggest keeping the Fig. 6 model in its original form.

4. Line 66: The provide references (9-14) do not support the statement that “an antigen designated R28.. is associated with puerperal sepsis”. Some support that

certain serotypes are associated with puerperal sepsis, some which can encode R28, but this connection is not as strongly established as stated. Why is there is no association of *emm2*, *emm48*, and *emm77* with puerperal sepsis, which the authors show in Fig 1e to carry R28? Lower expression (1F) does not appear likely, as there is no binding differences (1G). The importance of this specific mechanism in the development of disease may thus be somewhat oversold throughout.

We apologise for the mistake and confusion of incorrect reference numbers. Our sentence in line 66 (now lines 64-66) is: “Epidemiological studies have repeatedly indicated that an antigen designated R28 ^{7,8}, which has properties as an adhesin ⁹ and typically is found in *emm28* strains, is associated with *S. pyogenes* strains causing puerperal sepsis outbreaks ¹⁰⁻¹⁵.” It is very important to note that we have used the word “outbreaks” in this sentence. This is because R28+ *S. pyogenes* strains have been repeatedly associated with puerperal sepsis outbreaks and not with sporadic puerperal sepsis cases.

Concerning the association of *emm28*, *emm2*, *emm48* and *emm77* with the expression of R28, we agree that it is important to address this point, since it may appear puzzling to readers that puerperal sepsis outbreaks are caused almost exclusively by *emm28* strains. There may be several explanations for this preponderance of *emm28* strains. First, it is well known that *emm* types distributions vary widely over time, implying that *emm28* strains may have been more common than strains of other *emm* types when outbreaks of puerperal sepsis were studied. Second, studies of large strain collections have shown that R28 is present in all *emm28* strains but only in some *emm2* and *emm77* strains (Green et al., 2005, J Infect Dis. 192, 760-70, PMID: 16088825; Chochua et al., 2017, mBio 8, e01422-17, PMID: 28928212) (data are not available for *emm48* strains, which are very rare). Of note, *emm77* strains have caused at least one outbreak of puerperal sepsis (Benenson et al., 2015, Infect Control and Hospital Epidemiol., 36(12):1488-90. PMID: 26486196) Third, the gene for R28 is located on an MGE, which may be spreading from *emm28* strains to strains of other *emm* types, implying that the presence of R28 in non-*emm28* strains may be relatively recent (Sitkiewicz et al., 2011, BMC Microbiol 11:65, PMID: 21457552). Fourth, our data indicate that R28 expression is higher in *emm28* strains than in *emm2* and *emm77* strains, a feature that may favor the ability of *emm28* strains to cause puerperal fever (Fig. 1F). Thus, several factors may contribute to the preponderance of *emm28* among strains causing outbreaks of puerperal sepsis. These aspects are now considered in the Discussion

Pages 17-18 Lines 371-387 “While R28 is found in *S. pyogenes* strains of several *emm* types, viz. *emm28*, *emm2*, *emm48* and *emm77*, only *emm28* strains are strongly associated with outbreaks of puerperal sepsis ¹⁰⁻¹². However, *emm77* strains have been reported to cause at least one outbreak ⁵⁴. There are several potential explanations for the association of *emm28* strains with outbreaks of puerperal sepsis. First, this could be because *emm28* strains are more common than *emm2*, *emm77* and *emm48* strains, as suggested by recent data on *emm* type distributions ^{13,55}. Second, this could be because the gene for R28, *spr28*, is present in all *emm28* strains but not all *emm2*, *emm77* and *emm48* strains ^{13,23}, a fact that might be attributed to relatively recent acquisition of the mobile genetic element carrying *spr28* into these lineages ²⁴. Thirdly, our data indicate that R28 expression is higher in *emm28* strains compared to *emm2* and *emm77* strains. Finally, it could be that *emm28* strains have enhanced transmissibility or virulence properties. Some or all these factors may contribute to the preponderance of *emm28* among strains causing outbreaks of puerperal sepsis. It is also important to note that sporadic

cases of puerperal sepsis are commonly caused by strains of *S. pyogenes* that do not express R28¹¹. This may occur if such strains bypass the initial barriers to infection, although they lack R28, in which case the disease may be like that caused by R28-positive *S. pyogenes* strains. However, it seems reasonable to assume that a smaller infection dose is required for R28-positive strains, a factor that may favor the occurrence of outbreaks.”

5. The images of Fig 3C are of poor quality, and increased resolution and contrast are needed to visualize. Scale bars are absent. The procedure for this experiment is also very unclear; in the methods, it's stated that the cells are infected with (an unknown) amount of *S. pyogenes* for 24 h. However, no bacteria are apparent – it would be expected for the dish to be overgrown and everything dead at this point, by acidification of the media or more direct toxicity. Can the authors clarify how this was done? The mechanism is stated to be “CEACAM1-dependent” (line 792), but this is not shown. CEACAM1 knockout cells should be tested. In addition to testing CEACAM1 ^{-/-} cells, cells expressing ‘nonbinding’ CEACAMs of Fig 6 should be tested.

The images are made using light microscopy. We have already recognised that the images are difficult to visualize as demonstrated by our inclusion of “marker” bars representing pre- and post-closure epithelial cell fronts. A change in contrast did not improve visualization. Therefore, we believe our chosen approach is the most appropriate approach to improve visualization for the reader. We thank the reviewer for requesting more clarity on the methods. ME-180 cell monolayers were cultured, scratches were inflicted and photographs were collected. The monolayers were then infected with *S. pyogenes* (1×10^8 CFU); we have now restructured this sentence to improve clarity for the reader. The monolayers were subsequently washed after 24 hours to remove the cultured bacteria to be able to take a clear photograph of the scratch (24-hour timepoint). If we did not remove the bacteria, we would be unable to image the wound. The monolayers were subsequently cultured in media containing antibiotics for a further 24-hours prior to the collection of further photographs (48-hour timepoint). The fact that the wounds continued to repair over the 24-to-48-hour time period, indicates that the ME-180 cells were not directly killed by *S. pyogenes* infection. Photographs were taken on a light microscope at 10X magnification, we have now added this detail and a scale bar. We did try to modify the contrast of images when constructing the original Figure, but this did not improve visualization. Therefore, we added bars that represent 0- and 24-hours epithelial cell fronts. We believe this is the best way for the reader to visualize the data. We have now added a small key to make this clearer. We have modified the methods to improve clarity for the reader.

Page 26 Lines 564-574 “The cells were incubated at 37°C with 5% CO₂ for 1 hour after addition of 125 μL of McCoy’s 5A + 10% FCS to each well. Four photographs of each scratch, representing 0-hour timepoint (t₀), were collected using a Nikon D5100 Digital Camera. For infection experiments, tissue culture wells were infected with 1×10^8 mid-logarithmic phase *S. pyogenes* cultured in TH-Y and subsequently resuspended in McCoy’s 5A containing 10% FCS. After 24 hours incubation at 37°C with 5% CO₂, monolayers were washed three times to remove bacteria and supplemented with McCoy’s 5A media containing 10% FCS containing penicillin/streptomycin. Four photographs of each scratch, representing the 24-hour timepoint (t₁), were collected. The cell monolayers were cultured for a further 24 hours at 37°C with 5% CO₂, prior to collection of four photographs of each scratch, representing the

48-hour timepoint (t2). The mean distance of each scratch across the four photographs was measured using ImageJ.”

We agree with the reviewer that an exploration of the specificity of the R28-CEACAM1 interaction in driving the delayed wound healing phenotype would be a valuable addition to the manuscript. However, we disagree with the reviewer that this should be assessed using CEACAM1 $-/-$ cells. This is because it has already been established in the field that one of the functions of CEACAM1 in epithelial cells is to regulate wound repair (Le Blanc et al. 2011, *Wound Repair Regen* 19(6):745-52, PMID: 22092845; Hayashi et al. 2020 *Biochem Biophys Res Commun* 531:100734, PMID: 32025578). Therefore, CEACAM1 $-/-$ ME-180 cells do not serve as a suitable model to investigate specificity, as these cells will have a delayed response to wound healing compared to control cells.

We therefore performed two sets of additional experiments to further investigate the specificity of our findings in the wound scratching model. Firstly, we assessed the capacity of purified R28 to suppress the closure of wounds in the ME-180 cell monolayer. This demonstrated that spiking the cell culture media with 10 μ g/mL R28 protein, but not the closely related control Rib protein, significantly suppresses the repair of wounds inflicted in ME-180 cell monolayers at 24 hours (Fig 4D, see below). This data unequivocally demonstrates that R28 is sufficient to suppress cell migration and wound repair and does not require any additional bacterial factors. Secondly, we assessed the capacity of the R28-IgI3 interactions with CEACAM1 to delay wound closure. In these experiments, spiking the cell culture media with 10 μ g/mL of rR28-IgI3 protein significantly suppressed the repair of wounds inflicted in ME-180 cell monolayers at 24 hours (Fig 4E), consistent with the results obtained using purified R28. Importantly, rR28-IgI3 proteins with alanine mutations at residues I53 and Y61, that abrogate interaction with CEACAM1, did not suppress wound repair. As a control, we included the E49A mutant protein, which still binds CEACAM1, and yielded similar results as the IgI3 WT protein (Fig 4E, see below). This data reveals that the IgI3 domain of R28 suppresses cell migration and wound repair through its capacity to interact with CEACAM1.

Wound healing of ME-180 cell monolayers is impaired by the R28 protein. D) Wound healing of ME-180 cell monolayers upon challenge with purified R28 or purified Rib proteins. The mean \pm s.d. of $n = 8$ from four independent experiments. **E)** Wound healing of ME-180 cell monolayers upon challenge with rR28-IgI3 proteins. rR28-IgI3 and rR28-IgI3^{E49A} interact with CC1 (red circle), whilst R28-IgI3^{I53A} and R28-IgI3^{Y61A} do not interact with CC1 (open circle). The mean \pm s.d. of $n = 8$ from four independent experiments. **(These data panels are now integrated into Figures 4D and 4E)**

We titled the Figure “R28 promotes adherence of *S. pyogenes* to human cervical epithelial cells and suppresses wound closure through a CEACAM1-dependent mechanism”. Our data shows that R28 suppresses wound closure (Fig 4D and 4E) and that this is dependent on the interaction with CEACAM1 through the R28-IgI3 domain (Fig 4E).

6. In addition to testing CEACAM1 $-/-$ cells, cells expressing ‘nonbinding’ CEACAMs of Fig 6 should be tested.

ME-180 cells also express CEACAM1 in addition to CEACAM5 and CEACAM6 (Supplementary Fig. 7B and 7C) and (Swanson *et al.* 2001, Cell Microbiol 3(10): 681-91, PMID: 11580753).

Minor

7. Line 67-69: It is not clear over these lines how R28 is an important problem to solve for puerperal sepsis caused by other pathogens, which lack this protein.

We have refined our sentence to improve clarity for the reader

Page 4 Lines 68-71 “This problem is important given the global increase in puerperal sepsis cases caused by *S. pyogenes*^{2,16}, and the lack of vaccines against this pathogen. Moreover, insights about puerperal sepsis caused by *S. pyogenes* may provide information relevant to the pathogenesis of puerperal sepsis caused by other bacterial pathogens.”

8. Weckel et al (ref 19) is only cited that CEACAM modulates cellular activities (line 74) and anti-bacterial responses (177). However, this paper shows neither thing; it show R28 acts by binding multiple integrins. Line 295: Ref 20 regards HopQ, not R28. I suspect many of the references are just scrambled. The authors should confirm and correct these as appropriate. Its not possible to readily evaluate the overall scholarship as it is.

We apologise that some of the references were not in the correct order. We have now corrected this problem.

9. Fig 2D: it is not clear in the legend what a “panel of strains” could be refereeing to. Is this just wt and a mutant? Do *emm2*, *emm48*, *emm77* show a similar pattern? Naturally R28-deficient strains would be appropriate negative controls.

We thank the reviewer for noting that additional clarity was required here. We have modified the text.

Figure Legend 2D. “Binding of rCC1-HIS, rCC3-HIS, rCC5-HIS, rCC6-HIS and rCC8-HIS to a panel (n =15, including two isogenic $\Delta spr28$ strains) of *S. pyogenes* strains from lineage *emm28*”.

10. Fig 3B is somewhat unusual. The performance of the experiment is the same as 3A, but changed to a distorted linear scale. No statistical methods are provided here (or anywhere in the manuscript), but presumably these are also variable, or potentially incorrect. Can the authors confirm the proper presentation and statistical analysis for this distribution of data? Including the R28-mutant in 3B is an important control to show specificity.

We agree with the reviewer that the data in the old Fig 3A and Fig 3B can be more clearly presented and/or include additional controls.

We have repeated the experiments with the isogenic AL368 and AL368 $\Delta spr28$ strains and we present the data in a single Figure (Fig 4A) to improve clarity for the reader. The data demonstrate that blocking CEACAM1 with the monoclonal antibody but not isotype control reduces the adhesion of R28+ *S. pyogenes* to ME-180 cells. Data is now presented on a linear scale, and a description of statistical analysis is included in the Figure legend (One-way ANOVA). This data is presented in in Fig. 4A in the manuscript and in Fig A below.

We have also performed the experiments with an alternative isogenic (2369-97 and 2369-97 $\Delta spr28$) pair of strains. Importantly, the results for experiments using both isogenic pairs demonstrate that R28-expression is associated with enhanced adhesion of *S. pyogenes* to ME-180 cells. This data is presented in panel B below and in Supplementary Figure 7D.

Page 11 Lines 226-227 “Similar observations were made for an alternative isogenic pair of *S. pyogenes* strains (Supplementary Fig. 7d).”

Adhesion of *S. pyogenes* to ME-180 cell monolayers is promoted by R28 and requires CEACAM1 availability. Adherence of isogenic *S. pyogenes* A) AL368 strains and D) 2369-97 strains to ME-180 cells at a multiplicity of infection (MOI) of 10 (Data are represented as mean \pm s.d. of n = 5). Cells were pre-incubated with a monoclonal antibody (mAb) specific to human CC1 and CC5 N-terminal domains, as shown. The mean \pm s.d. of n = 10 from five independent experiments. **(These data panels are now integrated into Figure 4A and Supplementary Fig 7D.**

11. Line 294, reference needed

We have now rephrased this sentence and included references.

12. The ITC of Sup Fig 2 does not clearly the affinity for rCC1-N, unless that is the same as previously published, and the only new data all the non-binders?

The ITC data for binding of R28-IgI3 to CC1-N (now in Suppl. Fig. 4) is the same as previously published. We have further clarified this in the Figure legend. All other data displayed in this Figure is novel and reveals that other CEACAMs are non-binders for R28-IgI3.

Supp Fig 2 legend “Our previous ITC assay (left-most panel in A and B) revealed that R28-IgI3 and rCC1-N interaction and affinity of $K_D = 1050 \pm 18$ nM, ($\Delta H = -4.8 \pm 0.3$ kcal/mol), and is included here (van Sorge et al., 2020).”

Reviewer #2

1. In the manuscript „ The R28 adhesin of *Streptococcus pyogenes* promotes puerperal sepsis by targeting the CEACAM1 receptor “, Catton et al identify a novel CEACAM-binding adhesion from *Streptococcus pyogenes*, which not only seems to be the major (necessary and sufficient) adhesion driving pathogenesis of bacterial puerperal sepsis, but also interferes with epithelial wound healing, and contributes to subverting the innate immune response.

The manuscript is in general very well written, and the set up of experiments and their descriptions are adequate and can be easily followed. I have only a few comments for improvement, in general the manuscript requires only minor revision.

We thank the reviewer for these comments.

Minor points:

2. The effect on wound healing is very interesting. Is this mechanism specific for the R28 protein, or also observed with other bacteria derived CEACAM1 ligands, such as HopQ from *H. pylori*? Could this be included as a control?

We thank the reviewer for his/her interest in this problem. We agree that it will become interesting to analyze whether other bacterial CEACAM1-binding proteins interfere with wound healing, but studies of that problem are beyond the scope of the present paper. To support our observations that the mechanism of delayed wound closure is specific for R28, we have now performed additional experiments to demonstrate that supplementation of cell culture media with purified R28 or recombinant R28-IgI3 impairs the repair of the wounded ME-180 data. Importantly, using recombinant R28-IgI3 with mutations that abrogate CEACAM1 binding do not impact wound closure. Please see the response to Reviewer 1 comment 5.

3. The authors show that R28-IgI3 docks over the CEACAM cis/trans dimerization pocket, and some of the residues within the CEACAM-N domain were also shown to be involved in binding of other bacterial ligands, such as HopQ. In this context: Is CEACAM1 glycosylation involved in this interaction? While most bacterial adhesins binding to CEACAMs interact with sugar residues, HopQ is the only bacterial adhesion yet that was shown to bind via protein-protein interaction. Was this explored for R28?

All bacterial ligands identified to date target the centre of the C^{''}C'CFG face of CEACAM1. This face of CEACAM1 is characterised by the absence of a glycosylation site. Thus, it is very unlikely that CEACAM1 glycosylation is involved in the interaction of bacterial ligands, including R28, with CEACAM1. Our biochemical assays (such as Fig. 4, Supplementary Fig. 2 and Supplementary Fig.3) employed rCEACAM1-N expressed in *E. coli*, in which mammalian glycosylations are absent, provides evidence that CEACAM1 glycosylation is not required for binding R28-IgI3. This proves that the R28-IgI3 and CEACAM1 interact

via protein-protein interaction. Similarly, we previously demonstrated that the ligand from *S. agalactiae* β -Igl3 binds CEACAM1 through protein-protein interaction (PMID: 33522633).

4. R28 only binds to CEACAM1, while other bacterial adhesins bind to CEACAM 5 and 6 also. I am missing an interpretation of the binding interface and residues involved which helps to explain this differential behavior. Also, it would be interesting to model the interaction of the R28 protein with CEACAM1 in direct comparison to the interaction of HopQ and CEACAM1, which might help to understand the partially overlapping but seemingly also differential binding modes.

We agree that understanding the differential binding specificities of bacterial adhesins for CEACAM receptors is important for improving our knowledge of bacterial infection biology. The determinants of selectivity of other bacterial ligands are poorly understood. Formal biophysical assessments of R28 alongside other bacterial CEACAM1 ligands are required to understand determinants of selectivity, but such studies are beyond the scope of the present study. However, we have now added an interpretation of the R28 selectivity for CEACAM1 to the discussion, as suggested:

Pages 15-16 Lines 327-337 “The R28 adhesin exclusively binds to CEACAM1. In contrast, other bacterial adhesins bind to a combination of CEACAM1, CEACAM3, CEACAM5 and CEACAM6^{17,42-46}. Our finding therefore makes *S. pyogenes* the first pathogen for which CEACAM binding is restricted to CEACAM1. The basis of this selectivity requires further investigation and cannot be explained by targeting of R28 to different CEACAM faces, as R28 and all bacterial ligands characterized to date bind the A'GFCC'C" face^{17,42-45}. Moreover, the selectivity cannot be entirely explained by differential targeting of residues in the A'GFCC'C" face of CEACAM1, as residues F29, Q89, I91 and L95 on CEACAM1, which are targeted by R28, are also targeted by other bacterial adhesins including β protein of *S. agalactiae*, HopQ of *H. pylori* and Opa of *Neisseria. spp.*^{17,42,44,45}. The unique binding mode of R28 that is highly selective for CEACAM1 requires investigation using formal biophysical assessments.”

5. It was described that interaction of HopQ with CEACAM1 alters CEACAM cis/trans dimerization. Does R28 binding interfere with CEACAM cis/trans dimerization? Does it affect CEACAM1 dependent signaling?

We agree that these are interesting questions. Concerning the binding surface, we have now generated an additional figure (Suppl Fig. 5), based on our finding that R28 targets the centre of the C"C'CFG face of CEACAM1 (Fig 3A and 3D). As shown in Suppl. Fig. 5A and 5B, the R28-binding C"C'CFG face also forms the CEACAM1 trans-dimerization interface, implying that the binding of R28-Igl3 and other bacterial ligands interferes with CEACAM1 trans-dimerization and signalling. We have also performed an *in silico* analysis of CEACAM1 co-crystal structures to understand this competitive binding to CEACAM1. Superimposition of CEACAM1 with structures for bacterial ligands (R28-Igl3 of *S. pyogenes*, β -Igl3 of *S. agalactiae* or HopQ of *H. pylori*) reveals that the binding of all bacterial ligands is incompatible with CEACAM1 trans dimerization (Supplementary Fig 5C-E). We have included new text and the new Suppl. Fig 5 as shown below.

Pages 9-10 Lines 191-198 “As R28-Igl3 binds to the trans-dimerization interface (the A'GFCC'C" face) of CEACAM-N, we questioned whether the binding of R28-Igl3 to CEACAM1 could prevent CEACAM1

trans-dimerization interactions. Superimposition of the crystallographic CEACAM1 trans dimer (PDB identifiers 4WHD) with the R28-IgI3 complex shows that R28-IgI3 binding is incompatible with CEACAM1 trans dimerization (Supplementary Fig. 5a and b). Similarly, the binding of other bacterial ligands is incompatible with CEACAM1 trans dimerization (Supplementary Fig. 5c, d and e) ²⁶. These data suggest that infection with R28+ *S. pyogenes*, but not R28- *S. pyogenes*, would reduce cross-linking efficiency of CEACAM1.”

Whether the interaction between R28 and CEACAM1 affects signalling is an important problem, but it is beyond the scope of the present study.

R28-IgI3 binding disrupts CEACAM1 trans dimerization. **a** Superimposition of the crystallographic trans CEACAM1-N (CC1-N) dimer (PDB: 4WHD) with the (R28-IgI3)-(CC1-N) structure. **b** Surface structure of CC1-N showing critical residues for forming the trans CC1-N dimer or R28-IgI3 interaction. **c** Crystal structure

of CC1-N domain in complex with β -IgI3 (PDB: 6V3P). **d** Crystal structure of CC1-N domain in complex with HopQ (PDB: 6AW2). **e** Superimposition of the crystallographic trans CEACAM1-N (CC1-N) dimer with the (R28-IgI3)-(CC1-N) structure, (β -IgI3)-(CC1-N) and (HopQ)-(CC1-N) structure. **(These data panels are now integrated into Supplementary Figure 5)**

6. Discussion: The discussion is very short. immune regulating effects of other CEACAM binding pathogens should be compared. CEACAM1 selectivity is not discussed. Binding interface of R28 in comparison to other adhesins should be discussed.

We agree with the reviewer. We have now further discussed the following:-

- the immunoregulatory properties (see the response to reviewer 4 point 2)
- CEACAM1 selectivity and R28 interface (see the response to reviewer 2 point 4)
- the R28-IgI3 binding interface (see the response to reviewer 2 point 4)
- why R28-negative strains may also cause puerperal sepsis (see the response to review 1 point 3),
- why there is a preponderance of *emm28* strains with puerperal sepsis (see the response to reviewer 1 point 4).

7. R28 only binds to CEACAM1, while other bacterial adhesins bind to CEACAM 5 and 6 also. I am missing an interpretation of the binding interface and residues involved which helps to explain this differential behavior.

We agree with the reviewer. Please see the response to reviewer 2 point 4.

Reviewer #3

1. Puerperal sepsis is a devastating disease affecting many women around the world. The causative agent of puerperal sepsis is *Streptococcus pyogenes* (GAS), and the molecular mechanisms by which GAS is adapted to and able to cause puerperal sepsis is unknown.

Catton and Bonsor, et al. investigate the bacterial adhesin, R28, and its contribution to disease. The authors identified that R28 specifically binds to human CEACAM1, which is widely expressed in epithelial and immune cells. Through structural elucidation, the authors identify key residues in the binding pocket between R28 and CEACAM1, use blocking assays to prevent GAS binding to CEACAM1 and show decreased adhesion to epithelial cells. The authors identify R28-CEACAM1 interaction is required for suppressed immune response and decreased wound repair in vitro. These results are noteworthy and add mechanistic knowledge to the area of puerperal sepsis.

We thank the reviewer for these comments.

Major comments:

2. Statistical analyses were performed throughout the manuscript, yet the description of statistical analyses that were performed are not mentioned in figure legends or materials and methods.

We have now included descriptions of statistical analyses used across Figure legends. All changes are highlighted in yellow in the marked version of the revised manuscript.

3. Figure 1: The binding of R28 expressing strains to CC1 (Fig 1G) has a broad distribution for each *emm28*, *emm2* and *emm77* strains. I am curious regarding the R28 adhesion in the low binding strains: is the R28 sequence identical? Are there mutations present in the Igl3 like domain that could explain the decreased binding affinity?

We agree that the binding of CC1 to R28+ *S. pyogenes* isolates is variable. We have now typed our isolate collection for the presence and absence of the *spr28* gene by PCR. This data is now shown in Fig 1F, 1G and Supplementary Fig. 1G.

Page 6 Lines 111-113 "Of these, 28/28 *emm28*, 9/11 *emm2* and 8/10 *emm77* isolates carried *spr28*."

Taking the *spr28* PCR analysis and R28 expression data together, our data show that all *spr28*-positive strains express R28 at the bacterial surface.

Page 6 Lines 112-114 "Expression analysis (Supplementary Fig. 1e) showed that all strains carrying the *spr28* gene expressed R28, with highest expression in the *emm28* strains (Fig. 1f), which included a puerperal sepsis isolate from the 1930s."

The simplest explanation for the variation in CC1 binding is the difference in R28 expression. Indeed, we have demonstrated that R28 surface expression is variable between *S. pyogenes* isolates (Fig 1F), and it has been established in the literature that these differences are driven by mutations in an intergenic region that impacts *spr28* gene expression (Eraso *et al.* 2020 PLoS One 15(3):e0229064. PMID: 32214338). In agreement that R28 expression is a determinant of CC1 binding, surface detection of R28 and CC1-His binding was correlated (Supplementary Fig. 1G).

Page 6 Lines 117-120 “The variation in CEACAM1 binding is likely due to differences in R28 expression, as supported by a positive correlation (Supplementary Fig. 1g), due to mutations in an intergenic region that impacts *spr28* transcription levels ²⁵.”

We agree with the reviewer that low binding phenotypes of some strains could be attributed to the lack of the *spr28* gene or due to mutations in the Igl3 domain of R28. We have now analysed the R28-Igl3 sequence variation in the BLAST database which reveals that a single variable site in the Igl3 domain exists, corresponding to a V26A mutation. Residue 26 is not located in the CC1 binding interface. This provides good evidence that all, or the vast majority of, R28 proteins interact with CEACAM1. We have chosen to include this analysis in the structural section.

Page 9 Lines 184-190 “Finally, we investigated whether sequence variation in R28-Igl3 exists that could influence the capacity to interact with CEACAM1. The R28-Igl3 region was conserved in the genomes of those we tested for *spr28* positivity (Fig. 1e) and BLAST analysis against the entire NCBI *S. pyogenes* protein database confirmed a high level of conservation with just a single variant of valine to alanine at amino acid residue 26. This residue is not located in or near the CEACAM1 binding interface suggestive that all R28 proteins interact with CEACAM1.”

4. Figure 3: Fig 3A, a subset of the R28- replicates adhere just as well to the ME-180 cells as WT. Why do the authors think this is? Is there a second bacterial adhesin that could supplement the binding to ME-180 cells?

The data in Fig 3A is generated from 6 independent replicates. Variation in *S. pyogenes* adhesion is likely due to 1) differences in CEACAM1 expression on the cell monolayers between replicate experiments, 2) differences in R28 expression on *S. pyogenes* between replicate experiments, and/or 3) differences in the expression of other adhesin-receptor pairs between replicate experiments. However, the data consistently reveals enhanced adhesion of wildtype AL368 strain compared to Δ *spr28* strain to the ME-180 monolayers, as revealed in the spaghetti plot displayed in panel A below. We have now performed the analysis with an alternative pair of isogenic strains, as reported in the response to Reviewer 1, point 10. Again, the spaghetti plot displayed below (panel B) shows a clear difference between the two strains.

R28 expression enhances *S. pyogenes* adhesion to ME-180 cells. Adherence of isogenic *S. pyogenes* a) AL368 strains and b) 2369-97 to ME-180 cells at a multiplicity of infection (MOI) of 10, (Data are represented as mean \pm s.d. of $n = 5$) Statistical significance tested by one-way ANOVA with Tukey's post-hoc test (* $p < 0.05$, ** $p < 0.01$).

- The authors state that when GAS R28+ is present in the scratch assay that there is decrease wound healing. Is there host factors, such as EGF or associated proliferation pathways that the authors could use qPCR or Western blot to quantify the difference to further back up the claim of CEACAM1 binding is the only factor preventing wound healing?

We agree with the reviewer that it will be of much interest to perform a detailed molecular analysis of the mechanism, by which R28 interferes with wound healing, but such analysis is beyond the scope of the present study.

- Regarding immune suppression in human explants, would a similar response be observed if purified R28 was added to the explants or is the whole bacterial interaction with CEACAM1 required? A similar approach would be to incubate the irradiated bacteria with purified CEACAM1 and does the similar suppression occur?

We do not believe that adding purified R28 alone to tissue explants would induce immune suppression. This is because 1) an anti-bacterial immune response of tissue can only be measured in response to bacterial infection or in response to a bacteria-derived immunostimulant. Thus, if there is no immune stimulation, as would be the case with the addition of R28 alone, then there will be no immunosuppression. 2) CEACAM1 is an immunomodulatory co-receptor of many activating receptors including TLR2, TLR4, FcR, BCR and TCR. Co-receptor here means that binding of the CEACAM1 ligand alone does not trigger a cellular response, but in connection with the binding of another ligand (e.g. antigen such as LTA, LPS, IgG, etc) to its receptor (e.g. TLR2, TLR4, FcR etc.) leads to a modified reaction of the cells in contrast to cells only engaging the antigen. Thus, a

phenotype is only to be observed if a bacterial adhesin binds CEACAM1 and the other bacterial molecules/factors bind to corresponding activating receptors.

Minor comments

7. It is unclear which data set is displayed in Figure 4D. Is the neutrophil survival displayed occurring at 1 or 2 hrs? It would be beneficial to clarify if the data is a combination of both time points or a single time point is displayed.

We apologise for the confusion. We have updated the Figure legend and the methods

Figure Legend 5D “Survival of *S. pyogenes* after a 2-hour incubation with human neutrophils at a multiplicity of infection (MOI) of 10 was quantified as the percentage of inoculum.”

Page 28 Lines 617-618 “After 2 hours, 50 µL of sample was lysed with 0.3% saponin on ice. Lysed samples were serially diluted in PBS and plated onto TH-Y agar for CFU enumeration.”

8. Figure 4A model of cervical explant assay says cytokines were monitored at 24hrs after removal of GAS, which does not align with the material and methods section which mentions levels were monitored at 2 and 5 days. I find it curious that cytokine levels are monitored after removal of GAS, would it not be more representative of an infection for GAS to be present the whole time during the assay?

We thank the reviewer for raising the point that there was confusion in the description of these experiments. Our model was designed to measure the acute inflammatory response of cervical tissue to *S. pyogenes*. We harvested culture supernatants after the removal of *S. pyogenes* at 24 hours to allow detection of the early mucosal innate responses, but also to be able to detect differences in inflammatory markers induced upon infection by isogenic *S. pyogenes* strains. In this model there is an increasing baseline inflammatory response linked to the resection and processing of the surgical specimen. We selected the 48 hours post-incubation with stimuli time-point, because the inflammatory baseline is still low as shown in a previous publication (Herrera et al., 2021, Vaccines. 9(3):231, PMID: 33800213). Hence, the time point chosen for proteomic analysis allows the detection of changes in the cytokine/chemokine profile induced by incubation with *S. pyogenes* without masking by baseline inflammation.

We measured cytokine responses (via luminex assays) and tissue viability (via MTT assays) at 2 days and 5 days post-infection. In the manuscript, we only report data for 2 days post-infection as this was the time point in which we detected significant differences in the inflammatory response of the tissue to wildtype versus $\Delta spr28$ *S. pyogenes*. We thank the reviewer for pointing out that the text was confusing when we discussed a 5-day timepoint without presenting the data. To avoid confusion for the reader, we have a) modified the methods, b) modified Fig 5a to have “+24 hours” labels, c) removed the MTT data for the day 5 timepoint from Supplementary Fig 6b.

9. An interesting idea: are the authors considering developing a transgenic mouse line expressing human CC1 or a humanized mouse CC1? The alteration to the

Macaques CC1 shows promise that GAS R28 could bind other species CC1 if the protein was “humanized”.

Though humanized CEACAM1 mice do exist, no appropriate model of puerperal sepsis for such investigations do exist. Please see response to Reviewer 1 point 2. We have discussed this important issue in the discussion as below.

Page 18 Lines 388-398 “The most appropriate experiment to test the capacity of the R28-CEACAM1 interaction to promote development of puerperal sepsis would be to study disease progress in female mice, including transgenic human CEACAM1^{+/-} or ^{+/+}, after intravaginal inoculation with *S. pyogenes* shortly after the birth of pups. However, such a model that follows the natural route of puerperal sepsis does not exist. Given the large sacrifice of pups and the use of females who just had pups, it is highly unlikely that permission to perform such experiments would be provided by an ethical committee. Furthermore, colonization studies in transgenic mice, in the absence of birth-induced damage, are not justified as it is not representative of the fact that *S. pyogenes* only rarely colonizes the reproductive tract of women, even in late pregnancy⁵⁶. These arguments highlight the need to develop new models, that must be humanized given the human-specificity of the R28-CEACAM1 interaction, to investigate puerperal sepsis.”

Reviewer #4

1. This manuscript is a thorough and highly informative and impactful identification of a bacterial pathogen - host interaction. It was easy to follow and read, integrated many approaches to address the biological questions, and was of high quality in terms of data and presentation. I have a few comments and questions that would help a reader fully understand the results.

We thank the reviewer for these comments.

2. With respect to the neutrophil experiments, I am not sure what the authors think is going on. Are the bacteria only adhering to neutrophils and not being phagocytosed and that is why they aren't being killed? I believe both ceacam 1 and ceacam 3 binding can lead to phagocytosis. This is already a massive amount of data but this result does seem to need a bit more to be understood.

CEACAM1 is upregulated to the neutrophil surface when granules are released. Our data shows that purified R28 protein binds to the surface of activated human neutrophils, but not resting human neutrophils (Fig 5C), and that it exclusively binds CEACAM1 (Fig 2C, D and E). This indicates that R28, and likely R28-expressing *S. pyogenes*, can bind to activated neutrophils via CEACAM1.

Our neutrophil-*S. pyogenes* experiments demonstrate that R28-expression promotes *S. pyogenes* capacity to evade neutrophil killing (Fig 5D). We believe this indicates that R28 and CEACAM1 interactions subvert the neutrophil response against *S. pyogenes*. The mechanisms by which R28 could promote *S. pyogenes* survival in our assays are 1) R28 binding to CEACAM1 suppresses phagocytic uptake via alternative receptors (bacteria remain extracellular), or 2) CEACAM1 engagement is not associated with suppression of phagocytic uptake but is instead associated with enhanced survival of *S. pyogenes* in the phagolysosome (bacteria remain alive in phagolysosome). Studies of this mechanism are beyond the scope of the present paper

We have added clarity to these points in the results and discussion.

Page 14 Lines 292-294 "Of note, R28 does not interact with the other CEACAM receptors expressed by neutrophils (Fig. 2c, d and e), including the phagocytic CEACAM3 receptor^{37,38}. This suggests that R28 expressed on *S. pyogenes* will bind to activated neutrophils exclusively via CEACAM1." (Lines 291-293)

Pages 16-17 Lines 348-359 "*S. pyogenes* has evolved sophisticated mechanisms to evade neutrophil phagocytosis and killing⁵⁰. It is therefore of interest that activated neutrophils express not only CEACAM1, but also CEACAM3, a specialized receptor employed for phagocytosis and elimination of CEACAM3-binding bacteria^{37,38}. Since R28 does not bind CEACAM3, our data imply that CEACAM3 does not affect the role of R28 in the pathogenesis of *S. pyogenes* infections and puerperal sepsis. This lack of binding CEACAMs contrasts with most of the other bacterial ligands of CEACAMs, including Opa of *Neisseria spp.*^{44,45}, HopQ of *H. pylori*⁴², UspA1 of *M. catarrhalis*^{43,51}. This situation suggests that R28 has evolved to escape CEACAM3-mediated detection by neutrophils, whilst maintaining the capacity to interact with CEACAM1, allowing *S. pyogenes* to subvert their antibacterial responses. A better understanding of the intracellular signaling events affected by R28 engagement of CEACAM1 on

neutrophils could help to understand the progression of invasive infections and puerperal sepsis.” (Lines 347-358)

3. A few sentences comparing the binding site among the ceacams to report the molecular determinants about the ceacam selectivity would be of interest to a general reader. As well as indicating beyond Hop the other bacterial (and viral?) pathogens that engage ceacams. The scientists that work on these systems will find this interaction fascinating.

We agree with the reviewer. Please see the response to reviewer 2 comments 4 and 5.

Minor

4. I think there is a typo in the caption of Figure 1, I am not sure how immobilized R28-IgI3 would be used in a flow cytometry assay.

We have modified the text.

Figure Legend 1D. “Binding of HIS-tagged rCC1 variants to R28-IgI3, quantified by flow cytometry (Data are represented as mean \pm s.d. of n = 4)”

REVIEWERS' COMMENTS

Reviewer #1 (Remarks to the Author):

The authors have addressed all my prior concerns

Reviewer #2 (Remarks to the Author):

I thank the authors for addressing my questions as well as the points raised by the other reviewers. The manuscript has further improved, and I am confident that it is a major contribution to the field. I have no more questions.

Reviewer #3 (Remarks to the Author):

This resubmission by Catton and Bonsor et al is a significant contribution to the field of GAS and puerperal sepsis. The authors have done a thorough job in addressing this reviewer's concerns and have satisfied the issues raised.

Reviewer #4 (Remarks to the Author):

The authors have addressed my comments.